# Mapping and annotating genomic loci to prioritize genes and implicate distinct polygenic adaptations for skin color

Beomsu Kim [1,6], Dan Say Kim[1,6], Joong-Gon Shin[2,6], Sangseob Leem[2,6], Minyoung Cho [1], Hanji Kim[2], Ki-Nam Gu[2], Jung Yeon Seo [2], Seung Won You[2], Alicia R. Martin [3,4], Sun Gyoo Park[2], Yunkwan Kim [2], Choongwon Jeong [5], Nae Gyu Kang[2,7] ✉ & Hong-Hee Won[1,7] ✉

Evidence for adaptation of human skin color to regional ultraviolet radiation suggests shared and distinct genetic variants across populations. However, skin color evolution and genetics in East Asians are understudied. We quantified skin color in 48,433 East Asians using image analysis and identified associated genetic variants and potential causal genes for skin color as well as their polygenic interplay with sun exposure. This genome-wide association study (GWAS) identified 12 known and 11 previously unreported loci and SNP-based heritability was 23–24%. Potential causal genes were determined through the identification of nonsynonymous variants, colocalization with gene expression in skin tissues, and expression levels in melanocytes. Genomic loci associated with pigmentation in East Asians substantially diverged from European populations, and we detected signatures of polygenic adaptation. This large GWAS for objectively quantified skin color in an East Asian population improves understanding of the genetic architecture and polygenic adaptation of skin color and prioritizes potential causal genes.

Skin color is one of the few highly heritable phenotypes that varies between human populations because of strong selection for locally varying environments. At lower latitudes, where ultraviolet (UV) light is intense, dark skin protects against the photolysis of serum folate and has photoprotective properties[1,2], whereas at higher latitudes, light skin is advantageous for vitamin D synthesis in reduced UV-B light. This strong correlation between skin color and latitude is mirrored by signatures of positive selection around genetic variants that influence skin pigmentation[3], which may reflect local adaptation to regional UV environments[4,5]. The UV-B is responsible for vitamin D formation and affect endocrine gland functions and overall body homeostasis[6,7].

Genetic factors explaining skin color diversity according to ancestry and selection signals have been identified[4,8,9]. For example, mutations in genes known to affect the function of melanocytes, such as *MITF*, *MC1R*, *OCA2*, and *SLC45A2*, are the target of natural selection in a novel environment with reduced UV exposure at higher latitudes along human migration pathways from sub-Saharan Africa. Notably, Europeans and East Asians have both shared and unique signatures of positive selection related to skin pigmentation. For instance, they share the same allele of *KITLG* that causes light pigmentation; however, alleles of *SLC24A5* and *MC1R* are population-specific[4,9].

[1]Samsung Advanced Institute for Health Sciences and Technology (SAIHST), Sungkyunkwan University, Samsung Medical Center, Seoul 06351, Republic of Korea. [2]Research and Innovation Center, CTO, LG Household & Healthcare (LG H&H), Seoul 07795, Republic of Korea. [3]Analytic and Translational Genetics Unit, Department of Medicine, Massachusetts General Hospital and Harvard Medical School, Boston, MA 02114, USA. [4]Stanley Center for Psychiatric Research, Broad Institute, Cambridge, MA 02141, USA. [5]School of Biological Sciences, Seoul National University, Seoul 08826, Republic of Korea. [6]These authors contributed equally: Beomsu Kim, Dan Say Kim, Joong-Gon Shin, Sangseob Leem. [7]These authors jointly supervised this work: Nae Gyu Kang, Hong-Hee Won. ✉e-mail: ngkang@lghnh.com; wonhh@skku.edu

Recent genetic studies have identified substantial skin color diversity, even within the same population, and novel genetic determinants of skin pigmentation[10–12]. However, previous genetic studies on skin color-related traits have been conducted mostly in African and European populations, and the genetic architecture of skin color in East Asian populations remains poorly understood (Supplementary Data 1). Since the evolution of lighter skin pigmentation is among the most tantalizing examples of human adaptive evolution[13], a deeper investigation of the genetic determinants of skin color in East Asian populations is required to better understand the evolution of our own species.

Here, we conducted a large-scale genome-wide association study (GWAS) for objectively quantified skin color in 48,433 East Asians. We identified 23 loci associated with skin color, including 11 previously unreported loci, and showed the overall divergence of the identified genomic loci from the European population. Moreover, we quantified the interaction between genetic variants and sun exposure on skin color at the polygenic level. Our study provides further genetic evidence for the involvement of skin pigmentation genes in melanocytes and distinct polygenic adaptations under selection pressure worldwide.

## Results

### Study participants and quantification of skin color using image analysis

Of the 52,712 participants, 48,433 (91.9%) passed strict quality control procedures (see *Methods*). The skin color of the participants was quantified using the international commission on illumination (CIE) LAB values from photographs of sun-exposed skin: $L^*$, $a^*$, and $b^*$ values for skin luminance, red/green component, and yellow/blue component, respectively (Fig. 1). The distribution of the three skin color traits and other characteristics of the study participants was similar between camera resolution groups (group A and B skin color was measured using 18-megapixel images for 20,538 participants and 24.2-megapixel images for 27,895 participants, respectively) (Supplementary Data 2). $L^*$ and other skin color traits were negatively correlated ($L^*$ and $a^*$, Pearson's correlation coefficient $[\rho] = -0.58$, $P < 2 \times 10^{-16}$; $L^*$ and $b^*$, $\rho = -0.30$, $P < 2 \times 10^{-16}$), whereas $a^*$ and $b^*$ were positively correlated

($\rho = 0.32$, $P < 2 \times 10^{-16}$) in 40,790 unrelated participants. Age, sun exposure time per day, and duration of outdoor activity were negatively correlated with $L^*$ and positively correlated with $a^*$ and $b^*$ (Supplementary Fig. S1). Sunblock usage was positively correlated with $L^*$ ($\beta = 0.366$, $P < 2 \times 10^{-16}$) and $b^*$ ($\beta = 0.095$, $P = 3.02 \times 10^{-7}$) but negatively correlated with $a^*$ ($\beta = -0.203$, $P < 2 \times 10^{-16}$). The distribution and effect of sun exposure variables varied across age groups (young [<37 years], middle [37–49 years], and old [> 49 years] age groups) (Supplementary Fig. S2 and Supplementary Data 3). For example, the effect of sun exposure time per day and sunblock usage on $L^*$ was 1.50 (effect size of the interaction term between sun exposure variable and age group [$\beta_{sun \times age}$] = 0.203, $P = 1.22 \times 10^{-3}$) and 1.48 ($\beta_{sun \times age} = 0.155$, $P = 0.024$) times greater, respectively, in the old age group when compared with the young age group. Male participants had lower $L^*$ ($\beta = -4.848$, $P < 2 \times 10^{-16}$) and higher $a^*$ ($\beta = 2.566$, $P < 2 \times 10^{-16}$) and $b^*$ ($\beta = 1.005$, $P < 2 \times 10^{-16}$) values than female participants.

### GWAS of skin color

In the discovery phase, we performed a GWAS meta-analysis for each skin color trait ($L^*$, $a^*$, and $b^*$) in 48,433 East Asian participants (Supplementary Figs. S3 and S4). A total of 5,066,750 autosomal variants were tested for associations with skin color adjusted for age, sex, sun exposure variables, measurement month, genotyping array, and the first 10 principal components (PCs) of genetic ancestry (Fig. 2a). By applying a Bayesian linear mixed model (BOLT-LMM)[14] with PCs as covariates, no evidence of population stratification was observed in quantile-quantile plots of the GWAS results (Supplementary Fig. S5). The effect size estimates in the discovery GWAS were not correlated with the variant loadings of the first 10 principal components that represent the population structure (Supplementary Fig. S6). The genetic correlation across skin color traits exhibited the same trends as the phenotypic correlation ($L^*$ and $a^*$, $r_g = -0.72$, $P = 1.42 \times 10^{-43}$; $L^*$ and $b^*$, $r_g = -0.50$, $P = 9.48 \times 10^{-13}$; $a^*$ and $b^*$, $r_g = 0.21$, $P = 0.016$). A total of 138,839 variants on chromosome X were also tested for associations with skin color in 42,770 female participants. However, none of the tested variants exhibited statistical significance (Supplementary Fig. S7).

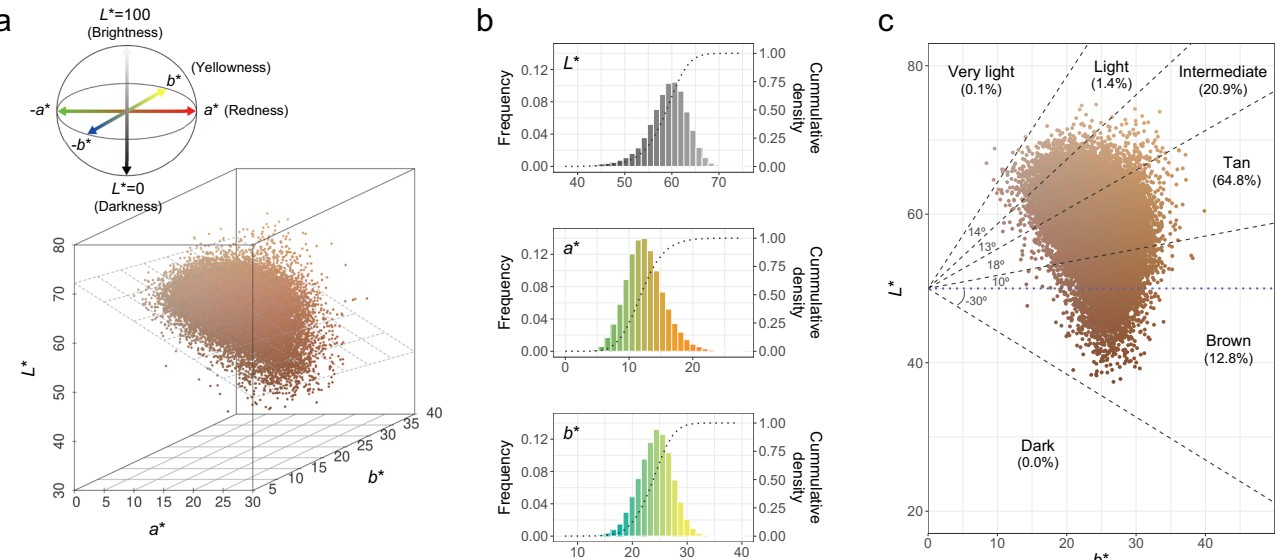

**Fig. 1 | Skin color distribution of participants. a** Three-dimensional distribution of quantitatively assessed skin color indices in the CIE LAB color space. Each dot corresponds to a study participant and its color represents the measured skin color for that person. The diagonal plane represents the regression plane for "$L^*$ - $a^*$ + $b^*$". **b** Distribution of $L^*$ (top), $a^*$ (middle), and $b^*$ (bottom). Histogram shows the frequency of each skin color trait, and dotted line represents the cumulative density.

**c** Distribution of categorical skin color classified by individual typology angle (ITA°) value: ITA° = [ArcTan(($L^*$ − 50)/$b^*$)] × (180/π). Each dot corresponds to a study participant and its color represents an average value of the measured skin color of both cheek areas for that person. Blue dotted line represents a horizontal line at $L^*$ = 50 and black dashed line represents the ITA° cutoff for categorical skin color.

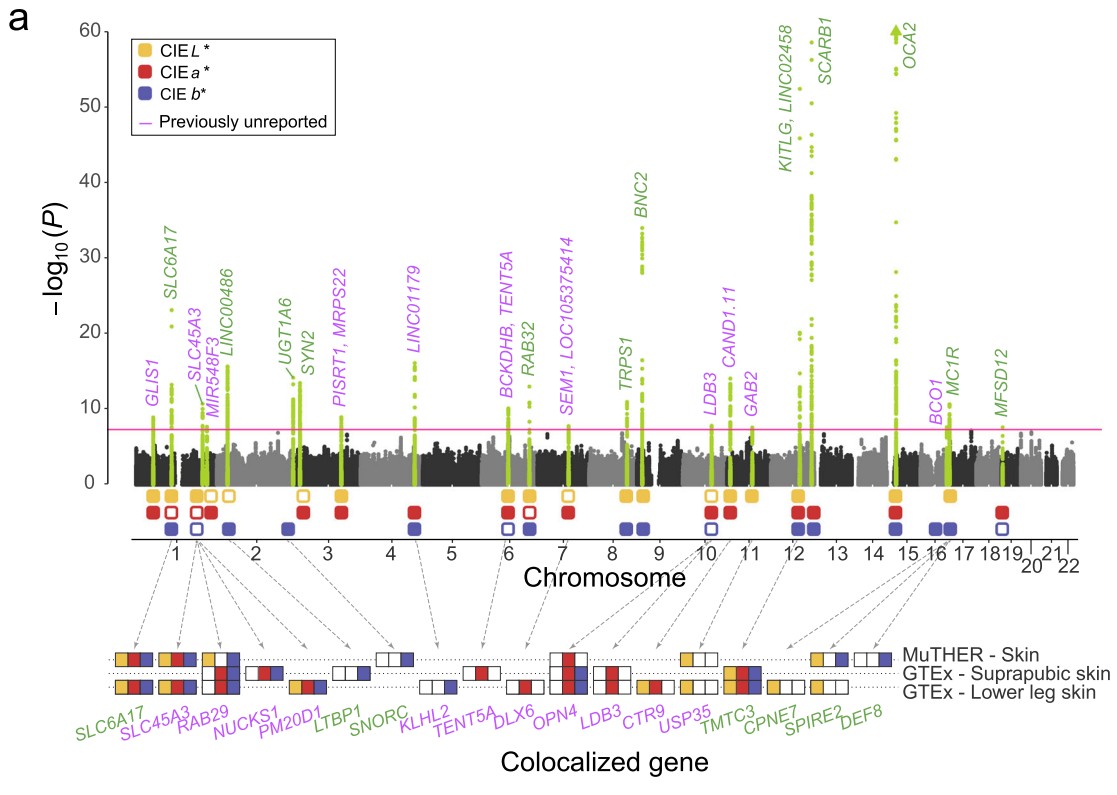

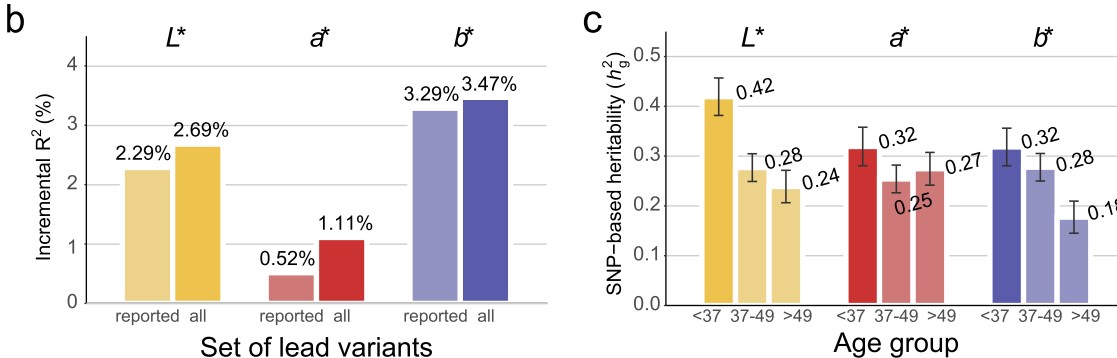

**Fig. 2 | GWAS of skin color traits with colocalization results and SNP-based heritability. a** Manhattan plot with $-\log_{10}(P)$ is presented for CIE LAB values of skin color, and genes colocalized in skin tissues are presented below the Manhattan plot. *P*-values were estimated using a two-sided score test in BOLT-LMM. The red horizontal line corresponds to the genome-wide significance threshold ($P = 5 \times 10^{-8}$). Genes in green and purple represent previously reported and unreported loci, respectively. Green dots indicate significant loci in at least one GWAS. Boxes in yellow, red, and blue represent significant loci of *L\**, *a\**, and *b\**, respectively; solid boxes indicate genome-wide significant loci and boxes with colored borderlines indicate nominally significant loci ($P < 2.17 \times 10^{-3}$, Bonferroni's

correction for 23 significant loci). For the boxes above colocalized genes, solid boxes indicate that a gene was colocalized (PP.H4 > 0.8) with GWAS of the color-corresponding phenotype. **b** Incremental $R^2$ value, defined as the increase in adjusted $R^2$ from the linear regression model relative to that from the model with covariates only. The incremental $R^2$ of lead SNPs on previously reported loci and on all identified loci are left and right for each trait, respectively. **c** SNP-based heritability by age group. For each skin color trait, SNP-based heritability of "young age" (<37 years, *N* = 11,369), "middle age" (37–49 years, *N* = 17,011), and "old age" (>49 years, *N* = 14,390) groups are described in order from left to right. Error bars indicate standard errors (SNP-based heritability estimates ± standard error).

We identified 26 lead variants at 23 independent loci associated with skin color traits, including 15 variants at 13 loci for *L\**, 15 variants at 13 loci for *a\**, and 13 variants at 12 loci for *b\**, respectively (Table 1 and Supplementary Data 4). Two independent lead variants were identified at each locus near *GLIS1*, *OCA2*, and *MC1R* (Supplementary Fig. S8). Among the 23 independent loci, 12 were previously reported and 11 were previously unreported, including 6, 8, and 2 loci for *L\**, *a\**, and *b\**, respectively. Detailed information on previous reports on skin color-related traits is provided in Supplementary Data 1. Among the previously unreported loci, *GLIS1*, *SEM1*, and *GAB2* have been identified as being associated with disease, particularly in inflammatory epidermal

conditions and melanoma[15–17]. This GWAS of objectively quantified skin color traits identified more significant loci than a GWAS based on the categorized skin color according to the individual typology angle (ITA°) value criteria (Fig. 1c and Supplementary Fig. S9). The GWAS of the categorical skin color using POLMM[18] identified 11 of 26 lead variants, with no additional significant loci, including two and nine variants in previously unreported and reported loci, respectively.

Including rs74653330 (p.Ala481Thr), a missense lead variant in *OCA2*, a total of 12 independent nonsynonymous variants associated with skin color traits were identified in the GWAS (Supplementary Data 5): 1 in a previously unreported locus, 9 in previously reported

**Table 1 | Lead variants associated with the skin color identified through meta-analysis of GWAS ($P < 5 \times 10^{-8}$)**

| rsID | GRCh37 | Traits | Nearest or colocalized genes | Function | A1 | A2 | L* β | L* s.e. | L* P | a* β | a* s.e. | a* P | b* β | b* s.e. | b* P | EAF |
|---|---|---|---|---|---|---|---|---|---|---|---|---|---|---|---|---|
| **Previously unreported loci** | | | | | | | | | | | | | | | | |
| rs7520584 | 1:54000726 | (L*),a* | GLIS1 | intronic | G | A | -0.029 | 0.007 | $1.93 \times 10^{-5}$ | 0.040 | 0.007 | $4.71 \times 10^{-9}$ | 0.008 | 0.007 | $2.18 \times 10^{-1}$ | 0.497 |
| rs702488 | 1:54204324 | L*,a* | GLIS1 | upstream | T | C | -0.045 | 0.008 | $3.36 \times 10^{-8}$ | 0.048 | 0.008 | $2.71 \times 10^{-9}$ | 0.015 | 0.008 | $6.01 \times 10^{-2}$ | 0.231 |
| rs7529037 | 1:205637501 | L*,(a*),(b*) | SLC45A3*, RAB29*, NUCKS1*, PM20D1* | intronic | G | C | 0.046 | 0.007 | $1.89 \times 10^{-11}$ | -0.032 | 0.007 | $3.22 \times 10^{-6}$ | -0.031 | 0.007 | $5.34 \times 10^{-6}$ | 0.457 |
| rs75998173 | 1:219110527 | (L*),a* | MIR548F3 | regulatory | G | A | 0.077 | 0.020 | $1.45 \times 10^{-4}$ | -0.113 | 0.020 | $2.16 \times 10^{-8}$ | -0.057 | 0.020 | $5.23 \times 10^{-3}$ | 0.029 |
| rs9853827 | 3:139002193 | L*,a* | PISRT1, MRPS22 | regulatory | T | A | 0.039 | 0.007 | $1.57 \times 10^{-8}$ | -0.042 | 0.007 | $1.14 \times 10^{-9}$ | -0.010 | 0.007 | $1.69 \times 10^{-1}$ | 0.409 |
| rs7667134 | 4:166643609 | a*,b* | LINC01179, KLHL2* | intronic | T | A | -0.016 | 0.007 | $1.86 \times 10^{-2}$ | 0.058 | 0.007 | $7.48 \times 10^{-17}$ | -0.041 | 0.007 | $3.02 \times 10^{-9}$ | 0.401 |
| rs9361843 | 6:82287485 | L*,a*,(b*) | BCKDHB, TENT5A* | intergenic | T | A | 0.038 | 0.007 | $3.68 \times 10^{-8}$ | -0.038 | 0.007 | $2.42 \times 10^{-8}$ | -0.023 | 0.007 | $8.78 \times 10^{-4}$ | 0.438 |
| rs6977057 | 7:96399494 | (L*),a* | SEM1, LOC105375414, DLX6* | intergenic | C | T | -0.039 | 0.007 | $5.01 \times 10^{-8}$ | 0.040 | 0.007 | $1.76 \times 10^{-8}$ | 0.000 | 0.007 | $9.78 \times 10^{-1}$ | 0.335 |
| rs3740342 | 10:88428718 | (L*),a*,(b*) | LDB3*, OPN4* | intronic | T | C | -0.024 | 0.007 | $5.57 \times 10^{-4}$ | 0.039 | 0.007 | $1.58 \times 10^{-8}$ | 0.023 | 0.007 | $9.07 \times 10^{-4}$ | 0.454 |
| rs2957668 | 11:10404382 | L*,a* | CAND1.11, CTR9* | intronic | T | C | -0.040 | 0.007 | $5.21 \times 10^{-9}$ | 0.052 | 0.007 | $4.66 \times 10^{-14}$ | -0.004 | 0.007 | $5.67 \times 10^{-1}$ | 0.483 |
| rs10899491 | 11:78106305 | L* | GAB2, USP35* | intronic | C | T | -0.039 | 0.007 | $2.73 \times 10^{-8}$ | 0.016 | 0.007 | $2.29 \times 10^{-2}$ | 0.006 | 0.007 | $3.62 \times 10^{-1}$ | 0.380 |
| rs924126 | 16:81283598 | b* | BCO1 | intronic | G | A | -0.007 | 0.007 | $3.36 \times 10^{-1}$ | 0.011 | 0.007 | $1.38 \times 10^{-1}$ | 0.041 | 0.007 | $2.49 \times 10^{-8}$ | 0.321 |
| **Previously reported loci** | | | | | | | | | | | | | | | | |
| rs6689641 | 1:110720400 | L*,(a*),b* | SLC6A17* | intronic | G | A | 0.053 | 0.007 | $1.65 \times 10^{-13}$ | -0.036 | 0.007 | $3.90 \times 10^{-7}$ | -0.072 | 0.007 | $7.11 \times 10^{-24}$ | 0.344 |
| rs10173066 | 2:33057305 | (L*),b* | LINC00486, LTBP1* | intronic | C | G | 0.029 | 0.007 | $2.56 \times 10^{-5}$ | -0.009 | 0.007 | $1.89 \times 10^{-1}$ | -0.057 | 0.007 | $2.10 \times 10^{-16}$ | 0.417 |
| rs10929285 | 2:234613677 | b* | UGT1A6, SNORC* | intronic | T | G | 0.008 | 0.007 | $2.44 \times 10^{-1}$ | 0.003 | 0.007 | $6.93 \times 10^{-1}$ | -0.054 | 0.007 | $6.35 \times 10^{-15}$ | 0.427 |
| rs3773364 | 3:12189968 | (L*),a* | SYN2 | intronic | G | A | -0.035 | 0.007 | $5.04 \times 10^{-7}$ | 0.052 | 0.007 | $3.32 \times 10^{-14}$ | 0.019 | 0.007 | $5.53 \times 10^{-3}$ | 0.455 |
| rs77310623 | 6:146871819 | L*,(a*),b* | RAB32 | intronic | A | G | 0.097 | 0.013 | $9.64 \times 10^{-14}$ | -0.054 | 0.013 | $3.86 \times 10^{-5}$ | -0.088 | 0.013 | $1.29 \times 10^{-11}$ | 0.074 |
| rs800884 | 8:116502628 | L*,b* | TRPS1 | intronic | C | A | 0.059 | 0.009 | $1.08 \times 10^{-11}$ | -0.012 | 0.009 | $1.56 \times 10^{-1}$ | 0.052 | 0.009 | $1.63 \times 10^{-9}$ | 0.193 |
| rs16935073 | 9:16795790 | L*,b* | BNC2 | intronic | C | A | 0.085 | 0.007 | $5.39 \times 10^{-34}$ | -0.019 | 0.007 | $7.29 \times 10^{-3}$ | -0.085 | 0.007 | $9.59 \times 10^{-35}$ | 0.423 |
| rs72620727 | 12:89110410 | L*,a*,b* | KITLG, LINC02458, TMTC3* | intergenic | C | A | 0.101 | 0.007 | $1.22 \times 10^{-46}$ | -0.048 | 0.007 | $1.31 \times 10^{-11}$ | -0.108 | 0.007 | $3.37 \times 10^{-53}$ | 0.375 |
| rs10846744 | 12:125312425 | a*,b* | SCARB1 | intronic | G | C | 0.021 | 0.007 | $3.73 \times 10^{-3}$ | -0.058 | 0.007 | $3.77 \times 10^{-15}$ | -0.119 | 0.007 | $2.48 \times 10^{-59}$ | 0.323 |
| rs7284404 | 15:28000042 | L*,a*,b* | OCA2 | intronic | C | T | -0.178 | 0.011 | $1.08 \times 10^{-62}$ | 0.061 | 0.011 | $7.78 \times 10^{-9}$ | 0.157 | 0.011 | $5.32 \times 10^{-50}$ | 0.117 |
| rs74653330 | 15:28228553 | L*,a*,b* | OCA2 | missense | T | C | 0.345 | 0.014 | $2.37 \times 10^{-137}$ | -0.127 | 0.014 | $5.12 \times 10^{-20}$ | -0.375 | 0.014 | $3.64 \times 10^{-162}$ | 0.065 |
| rs12598316 | 16:89916600 | L*,(b*) | SPIRE2*, DEF8*, CPNE7* | intronic | A | C | 0.059 | 0.010 | $5.06 \times 10^{-9}$ | -0.002 | 0.010 | $8.20 \times 10^{-1}$ | -0.052 | 0.010 | $2.95 \times 10^{-7}$ | 0.132 |
| rs33932559 | 16:89986025 | L*,b* | MC1R | missense | C | T | 0.118 | 0.019 | $6.39 \times 10^{-10}$ | -0.024 | 0.019 | $2.12 \times 10^{-1}$ | -0.126 | 0.019 | $3.66 \times 10^{-11}$ | 0.033 |
| rs2240751 | 19:3548231 | a*,(b*) | MFSD12 | missense | G | A | -0.017 | 0.007 | $1.95 \times 10^{-2}$ | 0.040 | 0.007 | $2.57 \times 10^{-8}$ | 0.030 | 0.007 | $4.12 \times 10^{-5}$ | 0.332 |

See Supplementary Data 1 for detailed information on previously reported variants. See Supplementary Data 4 for results in the replication GWAS and summary statistics of association between lead variants and raw phenotypes (without the inverse normal transformation). Colocalized genes in skin tissues are marked with asterisks. Abbreviations: GRCh37, chromosome number and base pair position (GRCh37/hg19); traits, genome-wide significant trait (nominally significant ($P < 0.05/23$ trait); A1, effective allele; A2, non-effective allele; β, coefficient of each SNP estimated by linear regression; s.e., standard error; EAF, effective allele frequency in the study sample; P, P-value of β.

loci, and 2 in nominally significant loci (top variants of 2 loci did not pass the genome-wide significance level, $P < 5.0 \times 10^{-8}$). Notably, a nonsynonymous variant, rs2511188 (p.Ile677Val), was identified in a previously unreported locus *USP35* ($P = 7.42 \times 10^{-7}$ for $L^*$), a gene that has been reported as a potential immunosuppressive factor in melanoma[19].

The explanatory power of the identified loci for skin color was measured by its incremental $R^2$ value, defined as the increase in adjusted $R^2$ from the linear regression model with covariates only to the model with covariates and lead variants (Fig. 2b). The incremental $R^2$ values of the lead variants in all significant loci, including previously unreported loci, were 2.69%, 1.11%, and 3.47% for $L^*$, $a^*$, and $b^*$, respectively. Compared to those of the lead variants in reported loci only, the incremental $R^2$ values of all loci, including previously unreported loci, were increased by 17.16%, 115.34%, and 5.57% for $L^*$, $a^*$, and $b^*$, respectively.

### Replication of GWAS results

The association between the lead variants and skin color traits was examined in 4,992 individuals (10.3% of the discovery cohort) who were externally independent of the discovery cohort (Supplementary Data 6). In the replication analysis, the effect sizes of the lead variants were comparable to those in the discovery GWAS (Spearman's correlation coefficients [$r_s$] between effect sizes: $L^*$, 0.908; $a^*$, 0.761; $b^*$, 0.832) and 8, 5, and 7 loci (lead variants or their proxies [LD $r^2 \geq 0.8$ and within a 50 kb]) showed nominal associations for $L^*$, $a^*$, and $b^*$, respectively ($P < 0.05$) (Supplementary Fig. S10 and Supplementary Data 4). The power-adjusted transferability (PAT) ratios of the discovery GWAS to the replication GWAS were 0.843, 0.552, and 0.729, for $L^*$, $a^*$, and $b^*$, respectively, which was calculated by dividing the observed number by the expected number of nominally significant loci (see *Methods*).

To assess the replicability under comparable sample sizes, we conducted 10-fold cross-validation on the discovery set (4843–4846 individuals in each validation set) (Supplementary Data 7). The effect sizes of the lead variants from the validation set were comparable to those from the training set in each fold: $r_s$ between effect sizes of the lead variants were 0.853–0.988, 0.659–0.882, 0.797–0.979 in GWAS for $L^*$, $a^*$, and $b^*$, respectively (Supplementary Fig. S11 and Supplementary Data 8). The PAT ratio in each fold of cross-validation was similar to that in the replication analysis: PAT ratios were 0.677–1.182, 0.552–1.000, 0.688–1.020 in GWAS for $L^*$, $a^*$, and $b^*$, respectively (Supplementary Fig. S12 and Supplementary Data 8). The limited replication of the lead variants, particularly for previously unreported variants, might be attributed to insufficient statistical power due to a small sample size. In the permutation meta-analysis of 10-fold groups (see *Methods*), the number of significant loci increased with larger sample sizes of GWAS, and previously unreported variants were not identified until the sample sizes reached 40–80% (60%, 40%, and 80% for $L^*$, $a^*$, $b^*$, respectively, based on the median number of significant loci) of the discovery GWAS (Supplementary Figs. S13, S14).

To assess the replicability of GWAS results at the polygenic level, polygenic scores for skin color in 4411 unrelated participants from the replication set were calculated using the discovery GWAS results for $L^*$, $a^*$, and $b^*$. Polygenic scores for $L^*$, $a^*$, and $b^*$ were significantly associated with the corresponding skin color traits ($L^*$, $\beta = 0.54$, $P = 7.79 \times 10^{-22}$; $a^*$, $\beta = 0.44$, $P = 1.11 \times 10^{-20}$; $b^*$, $\beta = 0.37$, $P = 1.89 \times 10^{-12}$) in each linear model adjusted for GWAS covariates (Supplementary Fig. S15).

### Heritability estimation and functional enrichment

Using linkage disequilibrium (LD)- and minor allele frequency (MAF)-stratified multicomponent genomic restricted maximum likelihood (GREML-LDMS) analysis, we estimated single-nucleotide polymorphism (SNP)-based heritability of skin color traits ($L^*$, $h_g^2 = 0.240$, s.e. = 0.010, $P = 7.12 \times 10^{-6}$; $a^*$, $h_g^2 = 0.238$, s.e. = 0.010, $P = 1.64 \times 10^{-9}$; $b^*$, $h_g^2 = 0.232$, s.e. = 0.010, $P = 6.29 \times 10^{-8}$). The highest LD score quartile (1st LD quartile) accounted for approximately half of the total SNP-based heritability (45–54%), and differences in SNP-based heritability across MAF quintiles were less than 0.02 (Supplementary Data 9). In consideration of the observation that the effect of the environment varied across age groups, we also estimated SNP-based heritability in the three female age groups (Fig. 2c and Supplementary Data 9). The SNP-based heritability of skin color traits was the highest in the "young age" group ($L^*$, $h_g^2 = 0.412$, s.e. = 0.038, $P = 1.77 \times 10^{-3}$; $a^*$, $h_g^2 = 0.310$, s.e. = 0.039, $P = 0.030$; $b^*$, $h_g^2 = 0.307$, s.e. = 0.038, $P = 0.012$).

Exocrine glands, skin, and connective tissue cells in the GWAS of $L^*$ (FDR = 0.089 for three tissues), and epithelial cells in the GWAS of $a^*$ (FDR = 0.001) were significantly enriched (Supplementary Fig. S16a and Supplementary Data 10) in tissue enrichment analysis using DEPICT[20]. The "penis foreskin melanocyte primary cells" in skin tissue had the highest odds ratio (OR) in the GWAS of $L^*$ (OR = 4.75, 95% confidence interval = [2.29, 9.89], $P = 3.00 \times 10^{-5}$) in the epigenetic feature enrichment analysis using GARFIELD[21] (Supplementary Fig. S16b).

### Colocalization with expression quantitative trait loci (eQTL) in skin tissues

To map potential causal genes, we performed a colocalization analysis between 23 genome-wide significant loci and eQTLs in 2 types of skin tissue (sun-exposed lower leg skin and non-sun-exposed suprapubic skin) from the Genotype-Tissue Expression project (GTEx) and skin tissue from the Multiple Tissue Human Expression Resource (MuTHER) using COLOC (Fig. 2a and Supplementary Data 11). The eQTL results of 18 genes from skin tissue were colocalized (posterior probability for colocalization [PP.H4] > 0.8) at genome-wide significant loci. *SLC6A17*, *SLC45A3*, *PM20D1*, *KLHL2*, *DLX6*, *CTR9*, *USP35*, *CPNE7*, and *SPIRE2* were colocalized in sun-exposed lower leg tissue, whereas *NUCKS1*, *LTBP1*, and *TENT5A* were colocalized in non-sun-exposed suprapubic tissue. In the 16q24.3 locus, which contains two lead variants in *SPIRE2* and *MC1R*, three genes were colocalized: *SPIRE2*, *DEF8*, and *CPNE7*. Although the function of *MC1R* in the skin is well known, the eQTLs of the colocalized genes, which showed a significant pattern corresponding to the GWAS results, were in LD with the lead variant in *SPIRE2* (Supplementary Fig. S17).

We also performed colocalization analysis with eQTLs in 47 other tissues from GTEx to investigate the possible pleiotropic effects of the identified loci (Supplementary Fig. S18). Among the genes colocalized in skin tissue, seven were also colocalized in other tissues, including *RAB29*, *OPN4*, *USP35*, and *SPIRE2* in more than three other tissues. Of the 50 tissues analyzed, the tissue with the most colocalized genes was the sun-exposed lower leg skin tissue from GTEx (Supplementary Fig. S19). Notably, in non-sun-exposed suprapubic skin tissue, only *TMTC3* near the *KITLG* locus was colocalized with the GWAS for $L^*$, whereas the tissue had the second most colocalized genes for $a^*$ and $b^*$.

### Single-cell level gene expression patterns

To identify cell type-specific gene expression patterns of skin color-associated genes, we analyzed two single-cell RNA sequencing (scRNA-seq) datasets of healthy skin tissues. To control for batch effects, we used the Harmony tool (v.1.0) and found that the same cell types from the datasets were well integrated (Supplementary Fig. S20a). The integrated scRNA-seq data were clustered into nine cell types based on the expression patterns of well-known cell type markers[22] (Fig. 3a and Supplementary Fig. S20b). In the integrated scRNA-seq data, 36 out of the 41 identified skin color-associated genes, including both the nearest and colocalized genes, were available.

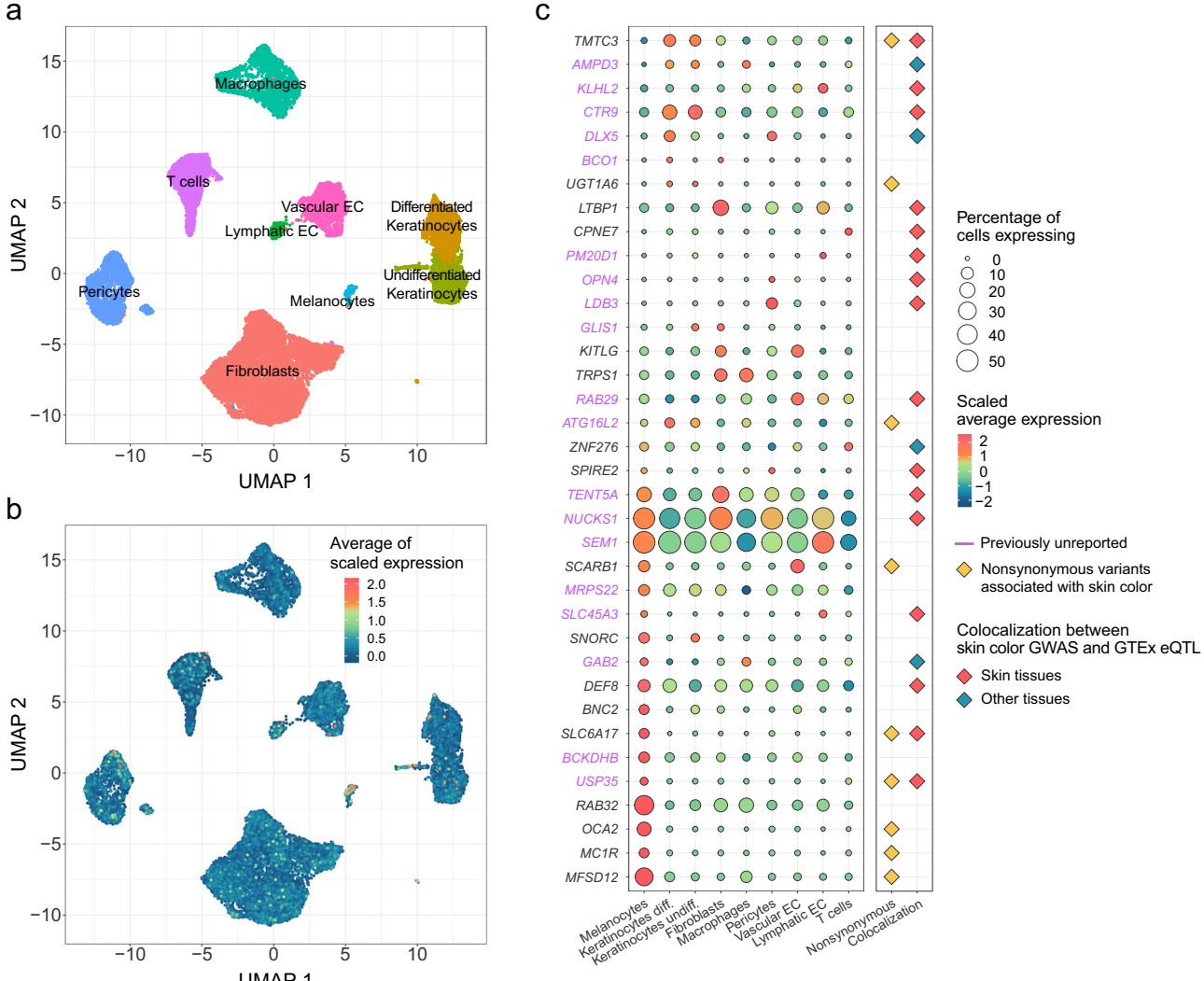

**Fig. 3 | Single-cell level gene expression patterns of CIE LAB values-associated genes. a** UMAP plot is shown for cell type identification. Cell types are identified with expression patterns of well-known cell type markers. Each color represents a cell type. **b** UMAP plot shows the overall expression patterns of CIE LAB-associated genes. Each cell is colored according to the average scaled expression of CIE LAB values-associated genes. **c** Dot plot of gene expression in cell types and additional evidence of each gene from the GWAS. Genes in purple represent those in previously unreported loci. The color of the circular dot represents the scaled average expression of each gene across cell types and size of the circular dot represents the percentage of cells expressing each gene within a particular cell type. Rhombic dots in yellow, red, and blue represent genes containing nonsynonymous variants associated with skin color, genes colocalized in skin tissue, or tissues other than skin, respectively. Abbreviations: Keratinocytes Diff. differentiated keratinocytes, Keratinocytes Undiff. undifferentiated keratinocytes, EC endothelial cells.

Skin color-associated genes showed relatively high RNA expression levels in melanocytes (Fig. 3b). At the individual gene level, one-third of the tested skin color-associated genes (12 of 36 genes) exhibited the highest expression levels in melanocytes (Fig. 3c and Supplementary Data 12), and 33% (4 of 12 genes; *USP35*, *BCKDHB*, *GAB2*, and *MRPS22*) were in previously unreported loci: *MFSD12*, *MC1R*, *OCA2*, *RAB32*, *SLC6A17*, *BNC2*, *DEF8*, and *SNORC* were in previously reported loci. Of the remaining 24 genes, 42% (10 genes) were highly expressed in fibroblasts (*NUCKS1*, *TENT5A*, *TRPS1*, *GLIS1*, *LTBP1*, and *BCO1*) and keratinocytes (*ATG1L2*, *UGT1A6*, *CTR9*, and *TMTC3*). Three genes, *NUCKS1*, *LTBP1*, and *TENT5A*, which were colocalized only in non-sun-exposed suprapubic tissue, were the most highly expressed in fibroblasts.

## Signatures of polygenic adaptation and association of genetic score with environmental factors

To identify evidence for the natural selection of the GWAS loci identified in the current study and compare the polygenic adaptation signal of those loci with the results of the skin color GWAS in UK Biobank (UKBB) Europeans, we calculated genetic scores for individual populations from the 1000 Genomes Project phase 3 and used the method proposed by Berg and Coop[23] to test for the overdispersion of genetic scores. Loci identified from the GWAS for $L^*$ and $b^*$ in the current study and the GWAS for light skin color in UKBB Europeans showed significant evidence for overdispersion of genetic scores globally ($L^*$, $Q_x = 59.82$, $P_Q = 3.90 \times 10^{-3}$; $a^*$, $Q_x = 38.87$, $P_Q = 0.052$; $b^*$, $Q_x = 47.88$, $P_Q = 0.020$; light skin color in UKBB Europeans, $Q_x = 357.62$, $P_Q = 1.53 \times 10^{-3}$) and significant polygenic adaptation in the corresponding regional population ($P$-value for regional population < 0.01) (Fig. 4a and Supplementary Fig. S21a). The GWAS loci for $L^*$ and $b^*$ identified in this study showed the highest genetic score and significant adaptation signal in East Asian populations, whereas those from the UKBB European GWAS showed the highest score and adaptation signal in European populations, implying that the genetic factors influencing skin lightness might vary across populations.

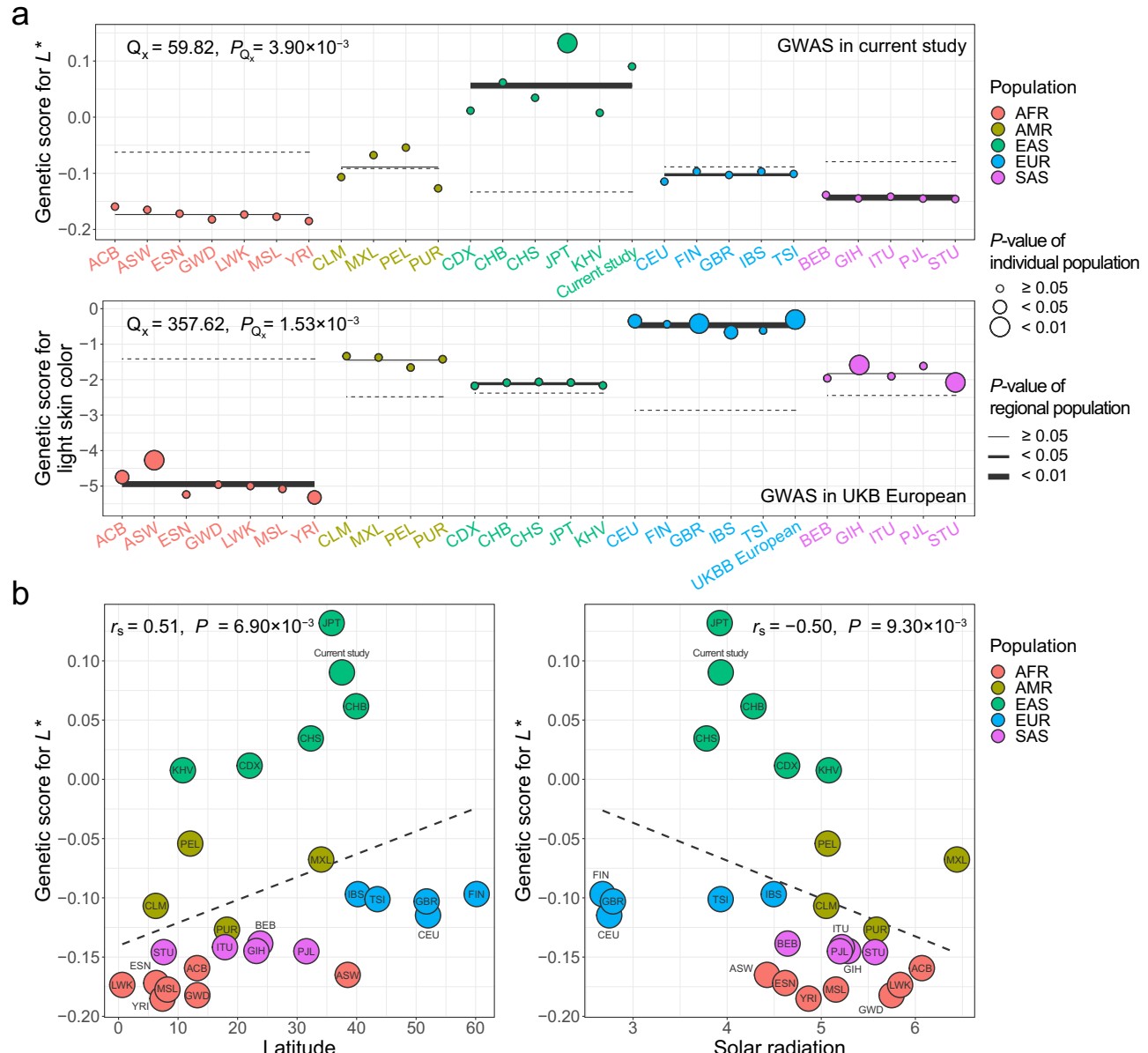

**Fig. 4 | Signals of polygenic adaptation for *L*\* across the 1000 Genomes Project phase 3 populations. a** Distribution of the estimated genetic score for *L*\* across the 1000 Genomes Project populations and results for polygenic adaptation based on the current GWAS (top) and the UK Biobank European GWAS (bottom). A test statistic for overdispersion of genetic scores ($Q_x$) and *P*-values are presented at the top of each plot (two-sided). **b** Estimated genetic scores for *L*\* based on the current GWAS are plotted against environmental factors: the absolute latitude of each population (left) and annual solar radiation (right). The regression lines (dashed lines) show the linearity between the genetic score (*y*-axis) and environmental factors (*x*-axis). Spearman's correlation ($r_s$) and *P*-values are presented at the top of each plot. The *P*-value of Spearman's correlation coefficient was estimated using a two-sided test under the null distribution of all possible permutations. Abbreviations: AFR African, AMR admixed American, EAS East Asian, EUR European, SAS South Asian, ACB African Caribbean in Barbados, ASW African Ancestry in South-west USA, ESN Esan in Nigeria, GWD Gambian in Western Division, Mandinka, LWK Luhya in Webuye, Kenya, MSL Mende in Sierra Leone, YRI Yoruba in Ibadan, Nigeria, CLM Colombian in Medellín, Colombia, MXL Mexican Ancestry in Los Angeles, CA, USA, PEL Peruvian in Lima, Peru, PUR Puerto Rican in Puerto Rico, CDX Chinese Dai in Xishuangbanna, China, CHB Han Chinese in Beijing, China, CHS Southern Han Chinese, China, JPT Japanese in Tokyo, Japan, KHV Kinh in Ho Chi Minh City, Vietnam, KOR Korean in the current study, CEU Utah residents with ancestry from Northern and Western Europe, FIN Finnish in Finland, GBR, British from England and Scotland, IBS Iberian Populations in Spain, TSI Toscani in Italy, UKBB European in the UK Biobank, BEB Bengali in Bangladesh, GIH Gujarati Indians in Houston, Texas, USA, ITU Indian Telugu in the UK, PJL Punjabi in Lahore, Pakistan, STU Sri Lankan Tamil in the UK.

Genetic scores for individual populations showed correlation with geographic and environmental factors, albeit with *P*-values estimated from the Mantel test ($P_{Mantel}$) generally being underpowered (Fig. 4b and Supplementary Fig. S21b). The absolute latitude and mean annual solar radiation by geographic region (see *Methods*) for individual populations from the 1000 Genomes Project phase 3 and their allele frequencies of GWAS lead variants are provided in Supplementary Data 13. The genetic score for *L*\* was positively and negatively correlated with absolute latitude ($r_s = 0.513$, $P_{permutation} = 6.90 \times 10^{-3}$, $P_{Mantel} = 0.160$) and mean annual solar radiation ($r_s = -0.496$, $P_{permutation} = 9.30 \times 10^{-3}$, $P_{Mantel} = 0.130$), respectively. Genetic scores for *a*\* and *b*\* were negatively and positively correlated with absolute latitude (*a*\*, $r_s = -0.581$, $P_{permutation} = 1.79 \times 10^{-3}$, $P_{Mantel} = 7.80 \times 10^{-3}$; *b*\*, $r_s = -0.562$, $P_{permutation} = 2.70 \times 10^{-3}$, $P_{Mantel} = 0.072$) and mean annual

solar radiation ($a*$, $r_s = 0.547$, $P_{permutation} = 3.63 \times 10^{-3}$, $P_{Mantel} = 0.121$; $b*$, $r_s = 0.526$, $P_{permutation} = 5.40 \times 10^{-3}$, $P_{Mantel} = 0.166$), respectively. Notably, the linear relationship between genetic scores and environmental factors was stronger among individual populations of East Asian ancestry ($L*$, $r_s$ with solar radiation ($r_s$-solar) $= -0.657$, $r_s$ with absolute latitude ($r_s$-latitude) $= 0.771$; $a*$, $r_s$-solar $= 0.771$, $r_s$-latitude $= -0.543$; $b*$, $r_s$-solar $= 0.600$, $r_s$-latitude $= -0.714$) than among populations outside of East Asia ($L*$, $r_s$-solar $= -0.313$, $r_s$-latitude $= 0.567$; $a*$, $r_s$-solar $= 0.418$, $r_s$-latitude $= -0.553$; $b*$, $r_s$-solar $= 0.357$, $r_s$-latitude $= -0.652$).

### Comparison of identified variants and polygenic score performance with the UK Biobank

We compared the genetic architecture of light skin color between East Asians and Europeans at the single variant and polygenic score levels using the GWAS results from the current study and the UKBB (Fig. 5). The skin color phenotype analyzed in the UKBB was limited to brightness, representing a categorical version of $L*$. In the UKBB East Asian sample ($n = 2332$), the effect sizes of the identified 15 $L*$ lead variants were comparable to those with the current GWAS ($r_s = 0.837$, $P = 2.77 \times 10^{-4}$), despite the lack of power. Although the sample size of the UKBB European participants ($n = 415,030$) was substantially larger than the sample size of our study ($n = 48,433$), the effect sizes of 720 lead variants from the UKBB European GWAS for skin color were less consistent compared to those in the current study ($r_s = 0.223$, $P = 7.34 \times 10^{-4}$) or the UKBB East Asian sample ($r_s = 0.111$, $P = 0.011$).

Polygenic scores for light skin color in the UKBB East Asian participants were calculated using GWAS results for $L*$ and light skin color from the current study and the UKBB European study, respectively. The Spearman's correlation coefficient ($r_s$) between the residual of polygenic score (adjusted for age, sex, and the first 10 PCs of genetic ancestry) and light skin color was approximately double when using the current GWAS results ($r_s = 0.090$, $P = 1.28 \times 10^{-3}$) than the UKBB European GWAS results ($r_s = 0.049$, $P = 0.076$). Polygenic scores derived using the current GWAS results differentiated between light and dark skin color better than when using the UKBB European GWAS results (Fig. 5b).

### Interplay between polygenic score and sun exposure for skin color

The polygenic scores of 40,790 unrelated study participants were derived from leave-one-out GWAS, in which target samples for score calculation were excluded via 10-fold partitioning (Supplementary Data 14). Sensitivity analyses of the polygenic score were performed using only female participants to remove the potential confounding effects of sex. To estimate the relative effects of the polygenic score and environmental factors on $L*$, the study participants were partitioned based on their polygenic scores for $L*$ and sun exposure variables (sun exposure hours per day and sunblock usage). The highest relative effect size for increasing $L*$ was observed in the group with a high polygenic score, low amount of sun exposure (less than 1 hour of sun exposure per day), and frequent sunblock usage, with the largest effect size attributable to the polygenic score (Fig. 6a). The increase in effect size due to sunblock usage was greater in the group with high sun exposure (more than 3 hours of sun exposure per day) than in the group with low sun exposure. The increase in relative effect size explained by the polygenic score was 1.37–2.69 and 1.52–4.46 times higher than the increase explained by sun exposure hours per day and sunblock usage, respectively.

To examine the interplay between polygenic factors and sun exposure on skin color, we evaluated the interaction effect between the polygenic score and sunblock usage for each group of sun exposure hours per day using linear regression models adjusted for covariates. The polygenic score for $L*$ and sunblock usage interacted negatively in the high sun exposure group (effect size of the interaction term [$\beta_{G\times E}$] $= -0.251$, $P_{G\times E} = 7.33 \times 10^{-3}$), but not in the low sun

exposure group ($\beta_{G\times E} = 0.038$, $P_{G\times E} = 0.452$). In environments with substantial sun exposure (more than 3 hours per day), the difference in predicted $L*$ by sunblock usage for study participants in the bottom 10th percentile of polygenic scores was more than twice as large as that for study participants in the top 10th percentile of polygenic scores (Fig. 6b). There was no significant interplay between the polygenic scores for $a*$ or $b*$ and sunblock usage, and no significant association between sunblock usage and $b*$ (Supplementary Fig. S22).

## Discussion

We aimed to objectively quantify skin color in 48,433 East Asians using image analysis and identify genetic variants and potential causal genes for skin color. This GWAS of objectively quantified skin color traits (CIE LAB values; $L*$, $a*$, and $b*$) produced more powerful results compared to GWAS based on ITA° value or questionnaire-based categorical skin color (Supplementary Data 1). We identified 23 skin color-associated loci, 11 of which were previously unreported, and the lead variants within the identified loci were examined in 4,992 individuals who were externally independent of the discovery cohort. The SNP-based heritability of skin color was estimated to be 24.0%, 23.8%, and 23.2% for $L*$, $a*$, and $b*$, respectively. The highest SNP-based heritability was estimated in the youngest age group (41.2%, 31.0%, and 30.7% for $L*$, $a*$, and $b*$, respectively, in females younger than 37 years), presumably due to their reduced environmental influence on skin color compared to other age groups. The explanatory power for skin color increased 1.06–2.15 times by identifying previously unreported loci. We identified twelve significant nonsynonymous variants associated with skin color. The genomic loci identified in this study were substantially divergent from those in the European population and a signature of polygenic adaptation was detected. The interaction between genetic variants and sun exposure on skin color was quantified at the polygenic level. Functional enrichment analyses showed significant enrichment of GWAS loci in primary skin and melanocyte cells. Potential causal genes for skin color were prioritized based on nonsynonymous variants, colocalization between GWAS and eQTL signals, and expression levels in melanocytes.

Skin color can be subdivided into brightness, redness, and yellowness, which can be represented by CIE LAB values, and is influenced by factors such as melanin, hemoglobin, oxyhemoglobin, and carotenoid levels[24]. Melanin serves as a primary factor in protecting the skin from UV radiation and its regulation within cells involves intricate mechanisms mediated by various endocrine and biochemical signaling pathways[25,26]. The skin color-associated loci identified in this study exhibited varying significance depending on the CIE LAB value, implying that these regions may be influenced by different factors and contribute to distinct biological pathways. For example, *MFSD12* contributes to red-yellow pigmentation by maintaining cysteine levels within melanosomes[27,28], and *SCARB1* influences skin yellowness by promoting the uptake of carotenoids[29,30]. Among the genes identified in this study, several have been reported to be functionally associated with melanin synthesis-related pathways: (1) *MC1R* in melanin synthesis signal transmission from keratinocytes to melanocytes[31]; (2) *TRPS1* and *KITLG* in the proliferation of epithelial cells and melanocytes[32,33]; (3) *OCA2* and *MFSD12* in the regulation of amino acid intake into melanosomes[28,34]; and (4) *KITLG*, *RAB32*, and *SPIRE2* in melanosome transport or dispersion within melanocytes[35,36].

Although their detailed functions in melanogenesis have not been fully elucidated, some genes have been reported to influence pigmentation-related traits. *UGT1A6*, which has a genetic effect on $b*$ (yellowness), is known to induce yellowish iris pigmentation[37] and is genetically associated with the clinical phenotype of Gilbert's syndrome (a common syndrome characterized by yellowish discoloration of the skin)[38]. In addition, the UGT1A gene family, including *UGT1A6*, is involved in the conjugation of vitamin D[39]. It is well known that UV-B exposure is a key factor in vitamin D synthesis[6]. In a previous study, the

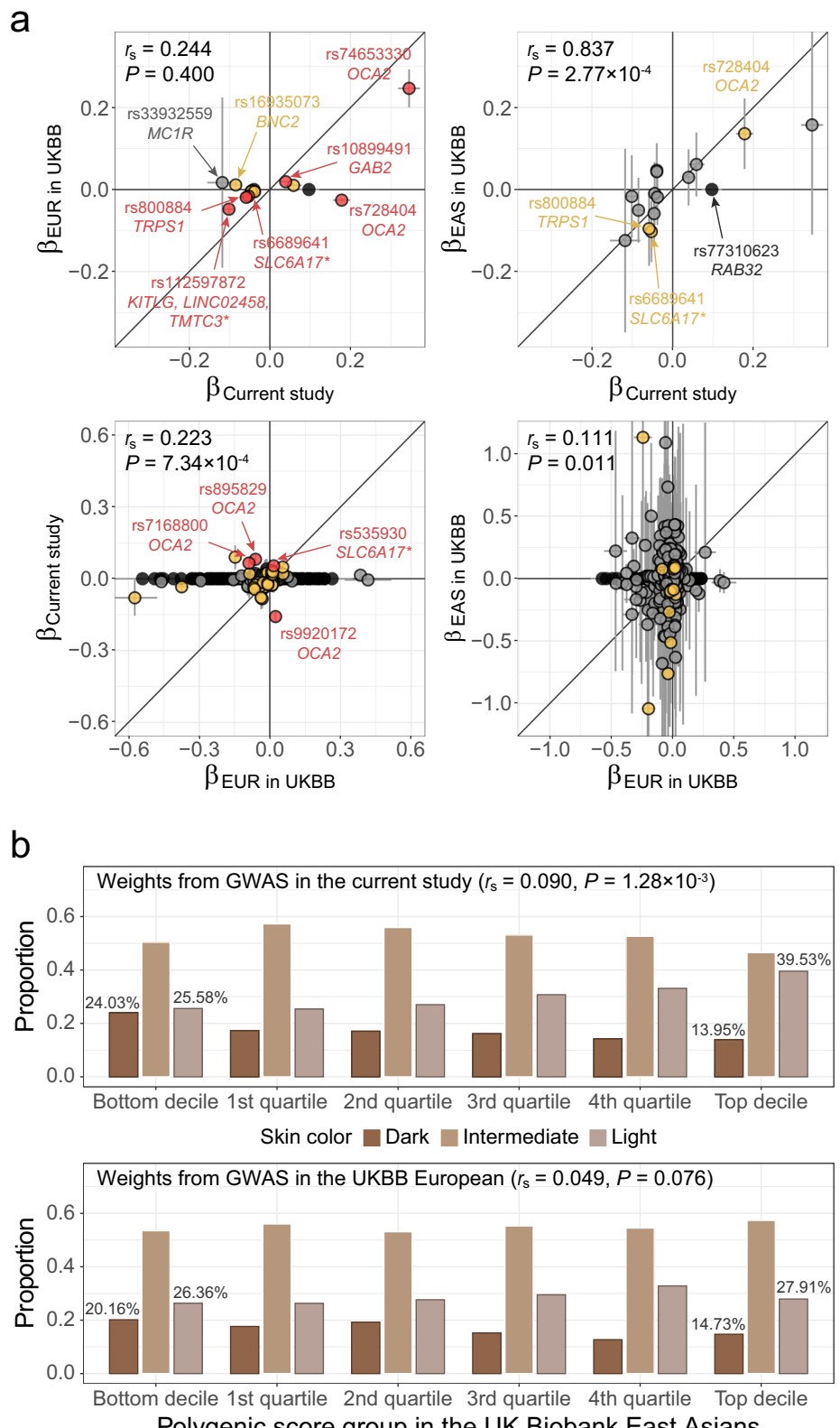

**b**

Weights from GWAS in the current study ($r_s$ = 0.090, $P$ = 1.28×10⁻³)

Skin color ■ Dark ■ Intermediate ■ Light

Weights from GWAS in the UKBB European ($r_s$ = 0.049, $P$ = 0.076)

Polygenic score group in the UK Biobank East Asians

membrane transporter gene *SLC6A17* was associated with pigmentation-related traits (tanning ability)[40].

Genes that have not been investigated for their functional role in pigmentation have been reported to be genetically associated with melanin- or melanocyte-related traits; for example, *TRPS1* is associated with tanning ability and skin cancer[41,42]. Most of the genes in previously reported loci and unreported loci were colocalized with eQTLs in skin tissues and were highly expressed in melanocytes, keratinocytes, and fibroblasts, which are known to play a role in repairing and remodeling the skin during the skin aging process[43]. A genetic variant of *SLC45A3* is associated with survival rate in melanoma and the function of the SLC45 gene family—mediating the transport of sugar molecules across the plasma membrane[44,45]. Despite skin color in East Asians being recognized as a phenotype with relatively low diversity compared to

**Fig. 5 | Comparison of lead variants for *L*\* and polygenic score performance with the UK Biobank. a** Comparison of lead variants for *L*\* from the current GWAS with the UK Biobank European GWAS (top left) and East Asian GWAS (top right), and comparison of lead variants for light skin from the UK Biobank European GWAS with the current GWAS (bottom left) and UK Biobank East Asian GWAS (bottom right). Dots in red, yellow, and gray represent genome-wide significant ($P < 5 \times 10^{-8}$), nominally significant ($P < 2.17 \times 10^{-3}$, Bonferroni's correction for 23 significant loci), and non-significant variants in the compared GWASs, respectively. Dots in black represent variants without results in the compared GWASs and are plotted along the *x*-axis. Colocalized genes in skin tissues are marked with an asterisk. Spearman's correlation ($r_s$) between effect sizes ($\beta$) of variants without black dots is presented at the top of each plot. **b** Distribution of polygenic score in the UK Biobank East Asian sample. The polygenic scores were calculated with weights from the current GWAS (top) and the UK Biobank European GWAS (bottom). For each decile or quartile of the polygenic score distribution, the proportion of participants who answered dark, intermediate, and light skin color is presented in order from left to right. Spearman's correlation ($r_s$) between the residual of polygenic score (adjusted for age, sex, and the first 10 PCs) and skin color and *P*-values are presented at the top of each plot. The *P*-value of Spearman's correlation coefficient was estimated using a two-sided test under the null distribution of all possible permutations.

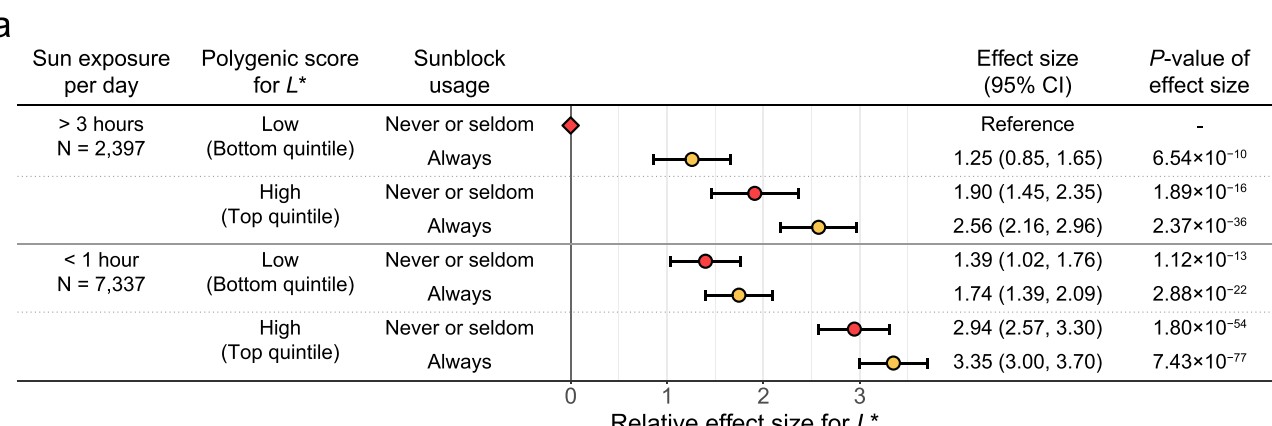

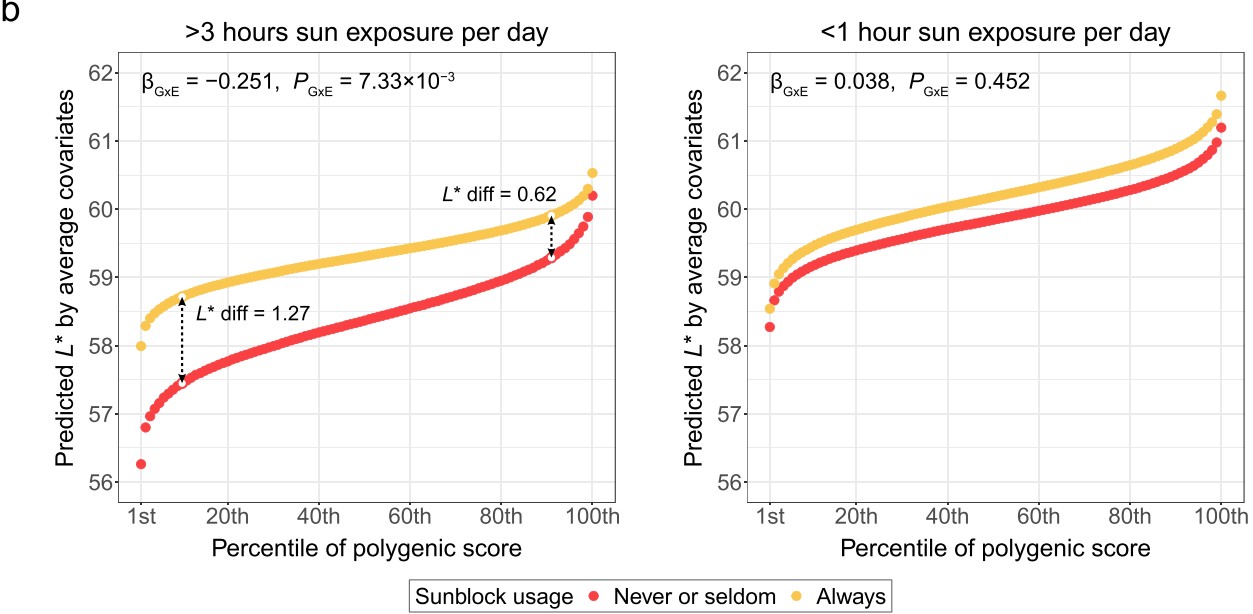

**Fig. 6 | Interplay of polygenic score and sun exposure for *L*\*. a** Relative effect size for *L*\* of each group divided according to sun exposure hours per day, polygenic score, and sunblock usage. A rhombic dot represents a reference group. Each dot represents the relative effect size and is colored according to sunblock usage. A total of 9734 independent individuals were examined in a linear model. Error bars indicate 95% confidence intervals for the relative effect size (relative effect size ± 95% confidence interval). **b** Predicted *L*\* by average covariates in each percentile of polygenic score distribution for participants with never or seldom sunblock usage (red) and always sunblock usage (yellow) within each group by sun exposure hours per day; more than 3 h (left) and less than 1 h (right). Effect size ($\beta_{G \times E}$) and *P*-values ($P_{G \times E}$) of interaction between polygenic score and sunblock usage within each sun exposure group are presented at the top of each plot. Abbreviations: CI confidence interval, *L*\* diff difference in *L*\* between sunblock usage groups at the top and bottom 10th percentiles of the polygenic score distribution.

populations worldwide, our study identified several potential causal loci for skin color in East Asians. These may serve as candidates for future functional studies investigating the molecular mechanisms underlying skin color.

The effect sizes of light skin color-associated lead variants in Europeans in the UKBB were generally less consistent with those in East Asians in the UKBB, whereas the effect sizes of the lead variants in our study had a linear relationship with those in East Asians in the

UKBB, even though the UKBB European sample size was more than eight times larger than that of our study. Consistently, the explanatory power of the polygenic score for light skin color in our study was approximately twice as high when applied to UKBB East Asians compared to the explanatory power of the score for UKBB Europeans. These findings are consistent with convergent evolution and a genetic architecture that differs from other widely investigated traits[46,47]. The polygenic adaptation signature of our GWAS loci was skewed towards East Asian populations. Among the 1000 Genomes Project phase 3 populations, the genetic score for $L^*$ was highest in East Asian populations, indicating that genetic factors influencing skin lightness might vary across populations, as also suggested previously[47]. The genetic scores for skin color traits were linearly associated with solar radiation and absolute latitude. Despite the relatively low diversity of these environmental factors in a regional population, the linear relationship between genetic scores and environmental factors was stronger in individual populations within East Asia than in those outside East Asia. In addition to the linear association between genetic and environmental factors, there was a significant interplay between these two factors. In environments with substantial sun exposure, participants with more alleles associated with darker skin color exhibited twice larger differences in skin lightness by sun exposure than participants with alleles associated with lighter skin color.

This study had certain limitations. First, most of the participants were women. The results of this study should be verified in the general population, and a study involving male participants is required to investigate sex-specific genetic factors. Second, although the phenotype had the advantage of being quantitative, it was measured in sun-exposed skin. To disentangle baseline skin color from tanning, the phenotype was adjusted for sun exposure-related covariates in the association test. Third, GWAS based on SNP array has limited ability to identify rare variants and population-specific signals. The utilization of population-specific reference panels for imputation might improve the discovery of additional loci[48]. Fourth, the association of the lead variants derived from the GWAS was partially replicated in 10% of individuals of the discovery cohort, although the effect sizes of the discovery result were consistent with the replication result. The identification of previously unreported loci associated with skin color in the current study might be attributable to the larger sample size than previous research in non-European populations. Fifth, the lack of resources for post-GWAS analysis, such as eQTL and scRNA-seq data generated from East Asians, may have resulted in insufficient power to detect evidence for the prioritization of potential causal genes. Sixth, to identify population-specific or shared genetic factors affecting skin color, GWAS in diverse populations and trans-ancestry meta-analyses are required. Finally, functional analyses of the identified genes are required to elucidate their underlying mechanisms.

Despite these limitations, our study has several strengths. This study included the substantial sample size in non-European populations and performed device-measured objective quantification of skin color, resulting in increased statistical power compared with previous skin color GWASs and the identification of many GWAS loci. Another strength of our study is that the GWAS signals were interpreted using various post-GWAS analyses using the latest statistical methods, along with transcriptome and single-cell resources. For example, well-known genes, including *SLC6A17*, *MFSD12*, *OCA2*, *UGT1A6*, *SCARB1*, and *MC1R*, as well as genes in previously unreported loci, including *TENT5A*, *SLC45A3*, and *SEM1*, have shown functional evidence in post-GWAS analyses. Our findings provide additional examples of population-specific skin color alleles that have accumulated independently in East Asian populations.

In summary, we conducted the large GWAS for objectively quantified skin color in an East Asian population and identified potential causal genes and polygenic adaptations based on diverse evidence. Our results provide fundamental information regarding the genetic architecture of skin color, which can further help elucidate the functional roles and molecular mechanisms of skin color-related genes in future studies.

## Methods

### Study participants

In the discovery phase of this study, 52,712 participants of East Asian ancestry who lived in South Korea and had no severe medical conditions at the time of recruitment were recruited through offline cosmetics shops in 2018. The following characteristics of all the study participants were evaluated: (1) measurement of facial skin color; (2) saliva collection for microarray genotyping; and (3) answering lifestyle questionnaires including age, sex, height, weight, disease history, average amount of sunlight exposure per day, and sunscreen usage. The Institutional Review Board (IRB) of the LG Household & Healthcare Research Center approved this study (IRB Nos. 2017-PB-0001 and 2018-PB-001). All study participants were fully informed of the study contents and signed written consent forms.

### Skin color measurement

To measure skin color, facial images were obtained using a Janus III system under normal light conditions (PIE Inc., Suwon, Korea). Participants were divided into two groups based on camera resolution: Group A, in which the skin color of 23,454 participants was measured using 18-megapixel images, and Group B, in which the skin color of 29,258 participants was measured using 24.2-megapixel images. Skin color image analysis was performed using an internal algorithm of the measuring instrument, which converted the images into numerical values. The following image analysis methods were applied to evaluate skin color: (1) RGB values of the analysis area for skin color, particularly for both cheeks (RGB scale range: 0–255); and (2) conversion of RGB values of each pixel into CIE LAB values: $L^*$ value for skin luminance (0 = dark, 100 = bright), $a^*$ value for the red/green component (positive value = red, negative value = green), and $b^*$ value for the yellow/blue component (positive value = yellow, negative value = blue).

### Genotyping, quality control, and imputation

To obtain reliable results from this study, we excluded participants with (1) measured images with low quality via manual curation and (2) the self-reported items that were possibly entered incorrectly (height below 1 m or above 2.5 m; weight below 30 kg or above 200 kg; age under 10 years). In total, 49,279 East Asians were genotyped.

Genotyping was conducted on DNA samples extracted from saliva using Illumina Global Screening Array MD BeadChips (Illumina, CA, USA). Sample- and variant-level quality control (QC) of the genotyped data was performed using an elaborate QC pipeline with PLINK (v.1.90)[49] (see Supplementary Notes for details). Thereafter, the genotype data were phased using Eagle (v.2.3)[50] and imputed based on the Haplotype Reference Consortium (r1.1, 2016) reference panel using Minimac (v.4)[51]. Genetic variants with low imputation quality scores ($R^2 < 0.8$) or MAF < 0.5% were excluded to reduce false-positive imputation results. In total, 48,433 East Asians were used for subsequent analysis in this study.

### Genome-wide association analyses

To assess the normality of residuals for each skin color trait (CIE LAB values; $L^*$, $a^*$, and $b^*$) in a null model (a linear model with only covariates), standardized residuals were compared to the standard normal distribution using the Kolmogorov-Smirnov test. All residuals were not normally distributed except for the residuals for $b^*$ in Group A (Supplementary Fig. S23). Accordingly, inverse-normal transformed residuals of each skin color trait after adjusting for age, sex, three sun-exposure variables, and measurement month were used for the GWAS in groups A and B. The sun exposure variables were questionnaire-based and classified into three categories (average sun exposure hours

per day, 1: more than 3 h, 2: 1–3 h, and 3: less than 1 h; sunblock usage, 1: never, 2: seldom, and 3: always; outdoor activities, 1: 3 or more times a week; 2: on the weekend; and 3: less than other categories). Associations between variants and skin color were tested using BOLT-LMM (v.2.3.4)[14], a Bayesian linear mixed model, to adjust for sample relationships. Association tests were conducted for groups A and B and adjusted for genotyping batches and the first 10 PCs of genetic ancestry. An inverse variance-weighted fixed-effects meta-analysis of skin color was performed to combine summary statistics from groups A and B using METAL (released on 2011-03-25)[52]. The results of the meta-analysis were double-genomic controlled. In the meta-analysis, variants that reached a genome-wide significance level ($P < 5.0 \times 10^{-8}$) were considered statistically significant.

To identify independent variants, a stepwise selection was conducted for each significant meta-analysis result using GCTA-COJO (v.1.91.2)[53]. The LD $r^2$ of these independent variants from all skin color traits was calculated and considered when selecting lead variants that represented significant loci of skin color traits; the LD $r^2$ across lead variants was less than 0.1. Lead variants were selected according to the priority of functional consequences (nonsense, missense, regulatory, intronic, or intergenic) and associated low $P$-value. The functional consequence and nearest gene were annotated using the Ensemble Variant Effect Predictor (VEP, v.98)[54]. Variants that passed Bonferroni's correction for 23 independent loci ($P < 2.17 \times 10^{-3}$) were considered as nominally significant and others were considered to have a null association with the skin color trait in the meta-analysis. To identify nonsynonymous variants, a Bonferroni's correction threshold for a total of 9183 independent nonsynonymous variants ($P < 5.44 \times 10^{-6}$) was used.

Genome-wide associations between variants and categorical skin color were tested using POLMM (released on 2022-08-26)[18], a proportional odds logistic mixed model. Skin color of the study participants was categorized with criteria of ITA° value[55], which has been used globally for skin color classification, to generate the categorical skin color. ITA° was calculated according to the following equation: $ITA_\circ = [ArcTan\left(\frac{L^* - 50}{b^*}\right)] \times \frac{180}{\pi}$.

To measure the explanatory power of the identified loci for skin color, the incremental $R^2$ value was estimated. Similar to ref. 56, the incremental $R^2$ value was defined as the increase in the adjusted $R^2$ from the linear regression model with covariates in the GWAS to the model with these covariates and lead variants.

## Replication of GWAS results

A total of 4,992 individuals of East Asian ancestry who lived in South Korea and had no severe medical conditions at the time of recruitment were recruited through offline cosmetics shops from 2020 to 2023 and participated for the replication of GWAS results. Genotyping, quality control, imputation, and association tests were performed following the same protocols as those applied to the discovery phase samples, except for the imputation quality scores ($R^2$): genetic variants with $R^2 < 0.6$ were excluded to maximize the number of replicated lead variants. Rs77310623, a lead variant in RAB32 for $L^*$ and $b^*$, was excluded from the replication study due to its absence in the imputed replication data.

The PAT ratio[57], determined by dividing the observed number by the expected number of nominally significant ($P < 0.05$) loci, was calculated to assess the replicability of the discovery GWAS. GWAS lead variants and their proxies (LD $r^2 \geq 0.8$) within a 50 kb window of each lead variant were selected to account for the observed number of loci. A locus was considered transferable if at least one of the tested variants was nominally significant and the direction of effect was consistent in both datasets. Power estimates were summed across the discovery GWAS loci for a given trait to provide an estimate of the number of loci expected to be significantly associated in the replication GWAS.

Participants in the discovery phase were randomly partitioned into ten groups to conduct cross-validation and permutation meta-

analysis, maintaining the proportion of females and camera resolution groups. Meta-analyses of all possible permutations of the ten groups, termed permutation meta-analysis, were performed to estimate the number of significant loci ($P < 5.0 \times 10^{-8}$) based on the discovery sample size.

## Heritability analysis

Genome-wide Complex Trait Analysis (GCTA, v.1.91.2beta)[58] was used to estimate SNP-based heritability of skin color traits, which is the proportion of variance explained by all SNPs. Notably, if causal variants have a different MAF spectrum than that used in the analysis or tend to be enriched in genomic regions with LD values higher or lower than the average, the estimated $h^2$ may be biased[59].

We used the GREML-LDMS[59] method in GCTA to estimate heritability with a region-specific LD heterogeneity correction. First, we calculated the LD scores of all SNPs using a sliding-window approach with a segment length of 200 kb (with a 100 kb overlap between two adjacent segments) and partitioned them into quartiles according to the LD scores. Second, each LD quartile was stratified into MAF quintiles with the same number of variants, resulting in 20 bins. Finally, we estimated the genetic relationship matrix (GRM) for each bin and jointly analyzed the 20 GRMs. We performed GREML-LDMS for skin color traits using age, sex, three sun exposure variables, measurement month, group, genotyping array, and the first 10 PCs as covariates.

To identify the age effect on the heritability of skin color traits in females, we divided the female participants into three subgroups: 11,369 participants younger than 37 years ("young age"), 17,011 participants aged between 37 and 49 years ("middle age"), and 14,390 participants older than 49 years ("old age"), and performed LD-stratified GREML analysis for each subgroup with the same covariates as the previous analysis. To evaluate the discrepancy of sun exposure variables and their effects on skin color across age groups, we conducted linear regressions in 36,246 independent female participants. The associations between age group and sun exposure variable were tested to assess the variation of sun exposure variable across age groups. The associations of the interplay between each sun exposure variable and age group on skin color were investigated to assess the discrepancy in the effects of sun exposure variable on skin color across age groups. These regression models included covariates such as measurement month, camera resolution, genotyping batches, and the first 10 PCs of genetic ancestry.

## Functional enrichment analysis

We performed functional enrichment analysis for the GWAS results with 36 categorized MeSH terms using DEPICT (v.1.1)[20]. Independent SNPs (LD $r^2 < 0.2$) that reached $P < 5.0 \times 10^{-4}$ were used for the DEPICT analysis. We considered significant enrichment after FDR correction for the 36 MeSH terms with $FDR < 0.1$.

We also performed tissue enrichment analysis using GARFIELD (v.2)[21] because it uses different databases, including ENCODE and Roadmap Epigenomics Projects. This tool performs greedy pruning with an LD parameter $r^2 > 0.1$ and calculates the overlap between LD-pruned variants and regulatory or functional annotations from the databases. ORs were calculated to quantify tissue enrichment at nine different levels ($T < 1$ to $T < 10^{-8}$) of GWAS $P$-value thresholds. Thresholds of enrichment $P$-values were applied as default settings.

## Colocalization with eQTL database

Colocalization between the GWAS results and eQTL was performed using the coloc.abf function in the coloc R package[60]. The eQTL results of 49 tissues, including 2 types of skin tissue (sun-exposed lower leg and non-sun-exposed suprapubic area) from GTEx v8[61] and skin tissue from MuTHER[62] were used for colocalization analysis. Colocalization with eQTL data was conducted for variants within 100 kb of each GWAS lead variant. For reliability, COLOC results that reached a high

posterior probability for colocalization (PP.H4) > 0.8, and eQTL *P*-value of the lead variant or its proxies (LD $r^2 \geq 0.8$) $< 1 \times 10^{-4}$ were considered colocalized. PP.H4 is the posterior probability for hypothesis H4, defined as a colocalized signal between a significant GWAS association and significant eQTL association.

## scRNA-seq analysis

We used two publicly available scRNA-seq datasets for scRNA-seq analysis. Both datasets were sequenced from healthy skin tissues using the 10× Genomics Chromium platform. One of the scRNA-seq datasets was downloaded from the Gene Expression Omnibus as a count matrix file (accession number GSE130973)[22]. The other scRNA-seq dataset was downloaded from https://dom.pitt.edu/wp-content/uploads/2018/10/Skin_6Control_rawUMI.zip as a raw UMI count table file[63].

Downstream analysis was performed using the "Seurat" R package (v.3.2.3)[64]. Initial filtering was performed using the following parameters: cells with fewer than 200 features or features detected in fewer than 3 cells. The two scRNA-seq datasets were combined using overlapping features that passed the initial filtering. For additional filtering after merging the datasets, we discarded cells with fewer than 500 features, more than 2000 features, or more than 5% mitochondrial content. A total of 24,203 cells and 19,321 features passed through all filters.

Data processing, including log normalization, selection of 2000 variable features using the variance-stabilizing transformation method, and data scaling was performed for visualization. Clustering was performed using the original Louvain algorithm. To reduce batch effects, the processed data were integrated using Harmony (v.1.0)[65] in the Seurat workflow. The integrated data were projected using UMAP[66] using the first 20 dimensions of the Harmony-corrected space. Cell types were identified using scaled average gene expression patterns of each cell type marker as described by ref. 65. Cell type markers are shown in Supplementary Fig. S16b.

We selected 49 skin color trait-associated genes, including colocalized genes, corresponding genes with nonsynonymous variants, and the nearest genes to the lead variants. Four of the selected genes were discarded during the scRNA-seq filtering process, and seven other genes were not included in the scRNA-seq datasets. The remaining 38 genes were included in the analysis.

## GWAS from the UKBB

For comparing the GWAS from the UKBB, summary statistics for skin color (data field 1717) were downloaded from the Pan-UK Biobank (https://pan.ukbb.broadinstitute.org/). Genome-wide significant variants ($P < 5.0 \times 10^{-8}$) with LD $r^2$ less than 0.1 with other variants were considered lead variants in the UKBB.

## Selection analysis

For analysis of the polygenic adaptation of lead variants, including nominal significance, from the GWAS summary statistics, we calculated genetic scores for individual populations from the 1000 Genomes Project phase 3[67] and used the method of Berg and Coop (released on 2014-12-21)[23] to test for overdispersion of genetic scores. The background selection values estimated by ref. 68 were used for this analysis. The genetic score of the $k$ individual population using $m$ GWAS lead variants is defined as $G_k = 2 \sum_{j=1}^{m} \beta_j p_{kj}$, where $\beta_j$ is an additive effect size estimate of the $j$ variant and $p_{kj}$ is an observed allele frequency of the $j$th variant in the $k$ individual population. The significance of the Q statistic, a statistic for overdispersion across populations, and *P*-value of individual or regional populations, which represents the divergence of the genetic values between the two populations relative to the null expectation under drift, were derived using GWAS lead variants from the current study and the UKBB European data. Independent variants (LD $r^2 < 0.1$) in both the GWAS and

1000 Genomes Project data were used to generate empirical null distributions for this test.

To evaluate the correlation between the genetic scores of lead variants and solar radiation, we used surface solar radiation data collected from January 1984 to December 2022 from the NASA POWER project (https://power.larc.nasa.gov/), and the mean annual solar radiation was calculated in kWh/m$^2$ per day units. Surface solar radiation values for each population were obtained by inserting the representative longitudes and latitudes (Supplementary Data 10). The longitude and latitude of each population were approximated based on the geographic region in which the population was investigated, similar to the study by ref. 10. The mean annual solar radiation and absolute latitudes were also used as parameters in the Berg and Coop analysis. Spearman's correlation coefficient ($r_s$) was used as a measure of correlation. To assess the significance of correlation, the *P*-value of $r_s$ was estimated under the null distribution of all possible permutations ($P_{permutation}$) and by Mantel test ($P_{Mantel}$).

## Polygenic score for skin color

Polygenic scores for skin color were calculated using the PRS-CS (released on 2021-06-04)[69] auto model, which has been reported to outperform other polygenic scoring methods for polygenic traits[70]. Polygenic scores of participants in the replication set, derived using the discovery GWAS, were used to assess the replicability of GWAS results at polygenic level. For 40,790 unrelated study participants, polygenic scores were derived from leave-one-out GWAS, where target samples for score calculation were excluded via 10-fold partitioning (Supplementary Data 11). To remove the possible confounding effects of sex, sensitivity analyses of the polygenic score with study participants were conducted using only female participants. We also calculated polygenic scores for the UKBB East Asian participants using the current GWAS results and the UKBB European GWAS results and compared the associations of polygenic scores with light skin color using Spearman's correlation coefficient ($r_s$).

## Interplay between polygenic score and sun exposure for skin color

To quantify the relative effect size of genetic and environmental factors on $L^*$, the study participants were divided into eight groups based on the top and bottom quintiles of polygenic score, sun exposure hours per day, and sunblock usage. The relative effect sizes of each group were calculated using multivariate linear regression adjusted for the covariates used in the GWAS (age, sex, measurement month, outdoor activities, genotyping batches, and the first 10 PCs of genetic ancestry).

The interaction effects between the polygenic score and sunblock usage were calculated for each sun exposure group using linear regression adjusted for the covariates used in the GWAS. The skin color values predicted by the average covariates were calculated by conditioning each polygenic score mean and covariates in each polygenic score percentile group.

## Reporting summary

Further information on research design is available in the Nature Portfolio Reporting Summary linked to this article.

## Data availability

The genotype and phenotype data of East Asian participants in the analysis were collected by Migenstory, a subsidiary of LG Household & Healthcare. This individual-level genotype and phenotype data are protected and are not available due to data privacy laws. The full summary statistics of GWAS for $L^*$ (luminance), $a^*$ (red/green component), and $b^*$ (yellow/blue component) in 48,433 East Asians are publicly available at the NHGRI-EBI GWAS Catalog (https://www.ebi.ac.

uk/gwas/downloads) with accession numbers GCST90320257, GCST90320258, and GCST90320259, respectively. The summary statistics of associations of variants in chromosome X with $L^*$ (luminance), $a^*$ (red/green component), and $b^*$ (yellow/blue component) in 42,770 East Asian females are publicly available at the NHGRI-EBI GWAS Catalog with accession numbers GCST90320260, GCST90320261, and GCST90320262, respectively. The UKBB genotype and epidemiologic data are available by requesting access on the UKBB homepage (https://www.ukbiobank.ac.uk/). The GTEx data are publicly available upon reasonable application (http://www.gtexportal.org/home/datasets). The MuTHER data are publicly available upon reasonable application (http://www.muther.ac.uk/Data.html). The scRNA-seq data were collected from the database in Gene Expression Omnibus (https://www.ncbi.nlm.nih.gov/geo/query/acc.cgi?acc=GSE130973) and University of Pittsburgh (https://dom.pitt.edu/wp-content/uploads/2018/10/Skin_6Control_rawUMI.zip). The surface solar radiation data from January 1984 to December 2022 were collected from the NASA POWER project (https://power.larc.nasa.gov/data-access-viewer). The 1000 Genomes Project phase 3 data are publicly available (https://www.internationalgenome.org/data).

## Code availability

Previously developed pipelines were used to produce the results for the current study. No custom code was developed. Please see the Supplementary Information for details on the software URLs and data used.

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

## Acknowledgements

This study was supported by LG Household & Healthcare (C17-03803 and C19-08213, N.G.K.). All data used in the analysis were collected by Migenstory, a subsidiary of LG Household & Healthcare. UKBB data were obtained under application no. 33002.

## Author contributions

Conceptualization: B.Kim, J.G.Shin, S.Leem, H.H.Won, N.G.Kang; Methodology: B.Kim, D.S.Kim, J.G.Shin, S.Leem, M.Cho; Formal analysis: B.Kim, D.S.Kim, J.G.Shin, S.Leem, M.Cho; Investigation: J.G.Shin, S.Leem, H.Kim, K.N.Gu, J.Y.Seo, S.W.You, A.R.Martin, Y.Kim; Resources: Y.Kim, S.G.Park, N.G.Kang, H.H.Won; Data curation: B.Kim, D.S.Kim, J.G.Shin, S.Leem, Y.Kim; Writing of original draft: B.Kim, D.S.Kim, J.G.Shin, S.Leem, M.Cho, H.H.Won; Writing, reviewing, and editing: B.Kim, D.S.Kim, J.G.Shin, S.Leem, H.Kim, K.N.Gu, J.Y.Seo, S.W.You, A.R.Martin, Y.Kim, C.Jeong, N.G.Kang, H.H.Won; Visualization: B.Kim, D.S.Kim, J.G.Shin, S.Leem, M.Cho; Supervision: N.G.Kang, H.H.Won; Funding acquisition: S.G.Park, N.G.Kang

## Competing interests

Migenstory's business is exclusively involved in providing Direct-to-Consumer (DTC) genetic testing services and generating data for research at LG H&H, without any engagement in the development of medicine or related technologies. J.G.S., S.L., H.K., K.N.G., S.W.Y., S.G.P., Y.K., and N.G.K. are employees of LG H&H. Other authors declare no other competing interests.
