## [Peer Review File · Nature Communications]

Mapping and Annotating Genomic Loci to Prioritize Genes and Implicate Distinct Polygenic Adaptations for Skin ColorReviewers' comments:

Reviewer #1 (Remarks to the Author):

This paper describes a relatively large GWAS of skin pigmentation in an East Asian population. There are many strong aspects to this paper – the phenotyping is well done, the analysis is largely technically appropriate (some comments on some of the analysis choices below). There is no substantial previous study of this phenotype in this population. I have some technical comments to improve the analysis below but overall I am highly enthusiastic about the paper and think that it is an important addition to the literature

General comments:

1) Conflict of interest. I see that the study was funded by LG and run by its subsidiary Migenstory, a precision medicine company. Several of the authors are employees. Are there patents, or commercial products associated or potentially associated with this research? Is there really no conflict? No problem but should be disclosed.

Comments about the GWAS:

2) Can we have a bit more information about the sample. Was this sample entirely recruited in Korea, or more broadly in East Asia – can we see a PCA plot of these samples projected onto East Asia. If it's actually about Koreans specifically this should be made clear in the abstract.

3) Why inverse normal transform the phenotype? That seems to be losing information and makes the betas uninterpretable in relation to other studies. Note in general there's no reason why the outcome variable in a linear regression needs to be normally distributed.

4) Why was the imputation performed with the HRC (which is largely European), rather than an East Asian reference panel (e.g. <https://www.science.org/doi/10.1126/sciadv.adg6319>) which should perform better?

5) There is no replication sample. That's unavoidable but a major limitation for the newly reported loci and should be mentioned.

6) Can you analyze the X chromosome? I understand the sample is mostly female (though I couldn't see a breakdown). I am sure there are some pigmentation genes on the X and also interesting in the light of sex differences in pigmentation that are mentioned.

Comments about the data:

7) I didn't see an accession number for the GWAS catalog results. Please make sure that's available. Are the phenotype and genetic data available? Please say if not.

Comments about other analyses

8) Residual stratification in the GWAS results can result in spurious Qx scores (Berg et al. eLife 2020, Sohail et al. eLife 2020), and could bias the selection results.

9) The P-values for the correlations between genetic score and latitude/solar radiation are not correct – the points are not independent so you cannot use the t-test (you need a mantel test or similar).

10) In Figure 4a there seem to be three sizes of circle on the plot, but only two described in the legend.

11) How did you decide the latitudes for the 1000 Genomes populations (STU & and ITU are from the UK, latitude ~50N and CEU are from Utah, latitude ~30). I get that you are trying to local the “ancestral” populations – please just explain how you did so.

12) I think it would be good to integrate the functional evidence from melanocytes in this study: <https://www.science.org/doi/10.1126/science.ade6289>. How many of your novel genes were identified here.

Reviewer #2 (Remarks to the Author):

This manuscript describes the results of a GWAS on skin colour in East Asians together with some additional types of data analyses on the discovered genetic loci. I agree with the authors that this is an interesting study because skin colour GWASs have not yet been carried out on the level of study population diversity that the global distribution of skin colour diversity would expect, and thus what is needed to genetically understand the full range of skin colour in humans, for which global populations need to be involved in GWAS. Study sample size is with 48,500 not very high, but acceptable given the scarcity of skin colour GWASs in Asians. Seeing image-based, continuous skin colour phenotype measures being used for GWAS is appreciated. This reviewer has two major conceptual concerns and some minor technical concerns as outlined in the following.

Major conceptual concern:

1) The most concerned limitation of this study is the lack of direct replication of the novel genetic loci the authors discovered in their GWAS. Although the authors compared the effect of their lead SNPs in their study population with those in East Asians from UKBB and report “comparable” effects, UKBB East Asians are with 2300 small in number and the reported correlation of the effects is with $r=0.83$ not very strong, thus providing limited indirect replication evidence. Moreover, the authors do not present any direct replication evidence of their novel SNPs, not in UKBB East Asians nor in any other East Asian population data. In general, direct replication of novel discovery findings is common standard in GWASs. Any reader would be reluctant to consider the newly presented skin colour loci as true genetic determinants of skin colour without direct replication evidence being presented. Moreover, this reviewer would expect that for a publication in a journal of the high reputation as this one, direct replication evidence would be a prime prerequisite for publishing GWASs.

Why did the authors not use UKBB East Asians for direct replication testing? Why did the authors not leave out from their discovery GWAS dataset some of their samples and use them for direct replication testing? Although the latter would represent internal replication testing as such samples come from the same pool, together with UKBB direct replication results from a relatively small sample set, the combination of both direct replication efforts would give a better idea on replication of their novel findings than the indirect analyses that allow limited conclusions they have performed thus far in UKBB.

2) South Koreans used here are not widely known as expressing large i.e., categorical skin colour differences as known from other Asian countries such as India. Thus, the reader wonders how the measured quantitative skin colour variation in the Koreans used here compares to that of populations with large categorical skin colour differences. If such comparative data are not available, it would be interesting to know how the measured quantitative skin colour variation translates to variation in commonly used skin colour categories. Are all Korean samples tested fall into one skin colour category or more categories? Such information may allow understanding why a sample size of close to 50,000 did not allow finding more than 23 associated genetic loci. Maybe the answer lies in the limited skin colour diversity of the study population, which also is a problem when carrying out skin colour GWASs in Europeans and in most samples collected within a single country that share similar genetic ancestry. The reported explained variance values make it clear that the number of skin colour genes in East Asians will be very high and most remain uncovered by this study. This may be due to their extremely

small effect size not detectable with the statistical power available with 50,000 samples, or by the low phenotypic diversity of the study population, or both. Shedding more light into this important aspect would be important for interpreting the authors' findings. Therefore, additional analyses on the phenotypic diversity of the study population would be highly appreciated.

Minor technical concerns:

- 3) The authors report different numbers of identified significantly associated genetic loci with the different quantitative measures of skin colour they used without discussing these differences. It would be interesting to know why, in the view of the authors, they obtained such different results.
- 4) Why was heritability testing done in an age-group stratified way? What is the authors' explanation that highest heritability was found in the youngest age group?
- 5) In the methods section it says that >52,000 samples were included in this study, while only 48,500 were used in the GWAS. How were the samples used for GWAS selected from the total samples? Why were 3,500 samples excluded?
- 6) The authors write that polygenic scores were estimated from 40,790 unrelated study participants. Does this imply that the 48,500 individuals used for discovery GWAS include 8000 relatives? I cannot see how such relatedness effects were adjusted for in the discovery GWAS.
- 7) What does "15, 15, and 13 variants at 13 loci for L*" in line 100 mean? Elsewhere in the manuscript, they refer to 15 L* lead variants.
- 8) The effect comparison with East Asian UKBB was done for 15 L* lead variants. Why only for these and not for all novel lead variants they discovered?
- 9) From their GWAS, the authors report 23 skin colour association genetic loci, while in their gene expression analysis they tested and provide results for 36 skin colour associated genes. How come the larger number of genes used in the gene expression analysis?
- 10) The UK Biobank study is typically abbreviated as UKBB, not as UKB used by the authors.

Responses from the Authors to Review Comments:

We thank the editor and reviewers for their constructive comments and suggestions. We have performed extensive analyses to address them. As a result, the Figures and Tables have been reordered due to the inclusion of additional supplementary materials and substantial modifications to the manuscript structure have been made.

Among the important and insightful comments provided by the reviewers, we have prioritized replication effort. Therefore, the results and discussion of the replication analyses are presented at the beginning of this response letter. We acknowledge the concerns raised regarding replication, and therefore, we have conducted an additional comprehensive replication study with 4,992 independent individuals for the replication phase (10.3% of the discovery cohort).

In the replication GWAS, the effect sizes of the lead variants were highly consistent with those in the discovery GWAS, although the significance level of the associations was limited (**Supplementary Fig. S10**). The power-adjusted transferability (PAT) ratio, determined by dividing the observed number by the expected number of nominally significant ($P < 0.05$) loci (Nature Communications, 2022), was calculated to assess the replicability of the discovery GWAS. The PAT ratios of the discovery GWAS for L^* , a^* , and b^* to the replication GWAS were 0.843, 0.552, and 0.729, respectively.

Supplementary Fig. S10.

To assess the replicability under comparable sample sizes, we conducted 10-fold cross-validation in the discovery set (4,843–4,846 individuals in a validation set) (**Supplementary Fig. S11**) as suggested by the reviewer. The PAT ratio of cross-validation was similar to that in the replication analysis: PAT ratios were 0.677–1.182, 0.552–1.000, 0.688–1.020 in GWAS for L^* , a^* , and b^* , respectively (**Supplementary Fig. S12**).

Supplementary Fig. S11.

a

Effect sizes from 10-fold cross validation of GWAS for L^*

b

Effect sizes from 10-fold cross validation of GWAS for a^*

c

Effect sizes from 10-fold cross validation of GWAS for b^*

◆ Novel ◆ Reported

Supplementary Fig. S12.

The limited replication of the lead variants, particularly for previously unreported variants, might be attributed to insufficient statistical power due to a small sample size. We conducted meta-analyses of all possible permutations of ten cross-validation groups to estimate the number of significant loci ($P < 5.0 \times 10^{-8}$) based on the discovery sample size. In the permutation meta-analysis, the number of significant loci increased with larger sample sizes of GWAS, and previously unreported variants were not identified until the sample sizes reached 40–80% of the discovery GWAS (Supplementary Figs. S13 and S14).

We also assessed the replicability of the discovery GWAS results at the polygenic level in 4,411 unrelated participants from the replication set. Polygenic scores for L^* , a^* , and b^* were significantly associated with the corresponding skin color trait (L^* , $\beta = 0.54$, $P = 7.79 \times 10^{-22}$; a^* , $\beta = 0.44$, $P = 1.11 \times 10^{-20}$; b^* , $r_g = 0.37$, $P = 1.89 \times 10^{-12}$) in each linear model adjusted for GWAS covariates (Supplementary Fig. S15).

Supplementary Fig. S13.

Supplementary Fig. S14.

Supplementary Fig. S15.

In summary, the association of the lead variants derived from the GWAS was partially replicated in the 10% of individuals of the discovery cohort, although the effect sizes of the discovery result were consistent with the replication result. The identification of previously unreported loci associated with skin color in the current study might be attributable to the larger sample size than previous research. Considering the limited replication in this study, we have revised ‘novel loci’ to ‘previously unreported loci’ throughout the manuscript, Figures, and Tables.

[Added to the Results, pages 6–7, lines 138–168]

Replication of GWAS results

The association between the lead variants and skin color traits was examined in 4,992 individuals (10.3% of the discovery cohort) who were externally independent of the discovery cohort (Supplementary Table

S6). In the replication analysis, the effect sizes of the lead variants were comparable to those in the discovery GWAS (Spearman's correlation coefficients [r_s] between effect sizes: L^* , 0.908; a^* , 0.761; b^* , 0.832) and 8, 5, and 7 loci (lead variants or their proxies [LD $r^2 \geq 0.8$ and within a 50kb]) showed nominal associations for L^* , a^* , and b^* , respectively ($P < 0.05$) (Supplementary Fig. S10 and Supplementary Table S4). The power-adjusted transferability (PAT) ratios of the discovery GWAS to the replication GWAS were 0.843, 0.552, and 0.729, for L^* , a^* , and b^* , respectively, which was calculated by dividing the observed number by the expected number of nominally significant loci (see **Methods**).

To assess the replicability under comparable sample sizes, we conducted 10-fold cross-validation in the discovery set (4,843–4,846 individuals in each validation set) (Supplementary Table S7). The effect sizes of the lead variants from the validation set were comparable to those from the training set in each fold: r_s between effect sizes of the lead variants were 0.853–0.988, 0.659–0.882, 0.797–0.979 in GWAS for L^* , a^* , and b^* , respectively (Supplementary Fig. S11 and Supplementary Table S8). The PAT ratio in each fold of cross-validation was similar to that in the replication analysis: PAT ratios were 0.677–1.182, 0.552–1.000, 0.688–1.020 in GWAS for L^* , a^* , and b^* , respectively (Supplementary Fig. S12 and Supplementary Table S8). The limited replication of the lead variants, particularly for previously unreported variants, might be attributed to insufficient statistical power due to a small sample size. In the permutation meta-analysis of 10-fold groups (see **Methods**), the number of significant loci increased with larger sample sizes of GWAS, and previously unreported variants were not identified until the sample sizes reached 40–80% (60%, 40%, and 80% for L^* , a^* , b^* , respectively, based on the median number of significant loci) of the discovery GWAS (Supplementary Figs. S13 and S14).

To assess the replicability of GWAS results at the polygenic level, polygenic scores for skin color in 4,411 unrelated participants from the replication set were calculated using the discovery GWAS results for L^* , a^* , and b^* . Polygenic scores for L^* , a^* , and b^* were significantly associated with the corresponding skin color trait (L^* , $\beta = 0.54$, $P = 7.79 \times 10^{-22}$; a^* , $\beta = 0.44$, $P = 1.11 \times 10^{-20}$; b^* , $r_g = 0.37$, $P = 1.89 \times 10^{-12}$) in each linear model adjusted for GWAS covariates (Supplementary Fig. S15).

[Added to the Discussion, page 12, lines 312–315]

We identified 23 skin color-associated loci, 11 of which were previously unreported, and the lead variants within the identified loci were examined in 4,992 individuals who were externally independent of the discovery cohort.

[Added to the Discussion, page 15, lines 384–388]

Fourth, the association of the lead variants derived from the GWAS was partially replicated in 10% of individuals of the discovery cohort, although the effect sizes of the discovery result were consistent with the replication result. The identification of previously unreported loci associated with skin color in the current study might be attributable to the larger sample size than previous research in non-European populations.

[Added to the Methods, page 19, lines 487–508]

Replication of GWAS results

A total of 4,992 individuals of East Asian ancestry who lived in South Korea and had no severe medical conditions at the time of recruitment were recruited through offline cosmetics shops from 2020 to 2023 and participated for the replication of GWAS results. Genotyping, quality control, imputation, and association tests were performed following the same protocols as those applied to the discovery phase samples, except for the imputation quality scores (R^2): genetic variants with $R^2 < 0.6$ were excluded to maximize the number of replicated lead variants. Rs77310623, a lead variant in *RAB32* for L^* and b^* , was excluded from the replication study due to its absence in the imputed replication data.

The PAT ratio⁵², determined by dividing the observed number by the expected number of nominally significant ($P < 0.05$) loci, was calculated to assess the replicability of the discovery GWAS. GWAS lead variants and their proxies ($LD\ r^2 \geq 0.8$) within a 50kb window of each lead variant were selected to account for the observed number of loci. A locus was considered transferable if at least one of the tested variants was nominally significant and the direction of effect was consistent in both datasets. Power estimates were summed across the discovery GWAS loci for a given trait to provide an estimate of the number of loci expected to be significantly associated in the replication GWAS.

Participants in the discovery phase were randomly partitioned into ten groups to conduct cross-validation and permutation meta-analysis, maintaining the proportion of females and camera resolution groups. Meta-analyses of all possible permutations of the ten groups, termed permutation meta-analysis, were performed to estimate the number of significant loci ($P < 5.0 \times 10^{-8}$) based on the discovery sample size.

We sincerely appreciate the time and effort that you have dedicated to reviewing our manuscript. Starting on the next page, we have responded to the reviewers' individual comment.

REVIEWER COMMENTS:

Reviewer #1:

■ Remarks to the Author:

This paper describes a relatively large GWAS of skin pigmentation in an East Asian population. There are many strong aspects to this paper – the phenotyping is well done, the analysis is largely technically appropriate (some comments on some of the analysis choices below). There is no substantial previous study of this phenotype in this population. I have some technical comments to improve the analysis below but overall I am highly enthusiastic about the paper and think that it is an important addition to the literature.

■ General comments:

1) Conflict of interest. I see that the study was funded by LG and run by its subsidiary Migenstory, a precision medicine company. Several of the authors are employees. Are there patents, or commercial products associated or potentially associated with this research? Is there really no conflict? No problem but should be disclosed.

Response: We appreciate the reviewer's positive evaluation of our work and important comments. We fully acknowledge the importance of disclosing potential conflicts of interest among authors. As mentioned in the Acknowledgements section, this study was primarily supported by LG H&H, while Migenstory's contribution was confined to the recruitment of participants and the collection of raw data prior to data analysis and writing of the paper. Migenstory's business is exclusively involved in providing Direct-to-Consumer (DTC) genetic testing services and generating data for research at LG H&H, without any engagement in the development of medicine or related technologies.

■ Comments about the GWAS:

2) Can we have a bit more information about the sample. Was this sample entirely recruited in Korea, or more broadly in East Asia – can we see a PCA plot of these samples projected onto East Asia. If it's actually about Koreans specifically this should be made clear in the abstract.

Response: We appreciate the reviewer's comment. Study participants who lived in South Korea and had no severe medical conditions at the time of recruitment were recruited. The reported ancestry ratio of the study participants is listed in **Supplementary Table S2**. Following the reviewer's suggestion, we have included **Supplementary Fig. S4**, which presents the principal component analysis (PCA) plot of the study participants projected onto East Asian from the 1000 Genomes Project phase 3. The PCA plot shows that the participants in the current study are positioned between Han Chinese and Japanese within the East Asians. Specifically, the Korean participants in this study were distributed between the Han Chinese (CHB and CHS) and Japanese (JPT), while the Chinese participants were closely positioned near the Han Chinese (CHB and CHS).

Supplementary Fig. S4

[Added to the Methods, page 16, lines 413–415]

In the discovery phase of this study, 52,712 participants of East Asian ancestry who lived in South Korea and had no severe medical conditions at the time of recruitment were recruited through offline cosmetics shops in 2018.

3) Why inverse normal transform the phenotype? That seems to be losing information and makes the betas uninterpretable in relation to other studies. Note in general there's no reason why the outcome variable in a linear regression needs to be normally distributed.

Response: We appreciate the reviewer's comment. The residuals for each skin color trait in a null model (a linear model with only covariates) were skewedly distributed. To examine the normality of residuals, we conducted a Kolmogorov–Smirnov test (**Supplementary Fig. S23**). The residuals for all skin color traits, except for residuals for b^* in the camera resolution group A, were significantly deviated from the normal distribution (P -value of the Kolmogorov–Smirnov test $< 0.05 / 6$, a threshold for Bonferroni's correction). To identify genetic variants using unbiased regression models while maintaining consistency between association tests, we performed a GWAS on the inverse normal transformed phenotype. In this study, post-GWAS analyses using polygenic score were performed on the original, untransformed phenotypes. Following the reviewer's suggestion, we have presented summary statistics from the GWAS without the inverse normal transformation in **Supplementary Table S4**.

Supplementary Fig. S23

[Added to the Methods, page 17, lines 449–452]

To assess the normality of residuals for each skin color trait (CIE LAB values; L^* , a^* , and b^*) in a null model (a linear model with only covariates), standardized residuals were compared to the standard normal distribution using the Kolmogorov-Smirnov test. All residuals were not normally distributed except for the residuals for b^* in Group A (Supplementary Fig. S23).

4) Why was the imputation performed with the HRC (which is largely European), rather than an East Asian reference panel (e.g. <https://www.science.org/doi/10.1126/sciadv.adg6319>) which should perform better?

Response: We appreciate the reviewer's recommendation. The reference panel of the study referred to by the reviewer (NARD2, 14,393 individuals) was released in December 2023, and at the time of the initial manuscript submission, only the reference panel of the previous version (NARD, 1,779 individuals) was available. In contrast, the Human Reference Consortium (HRC) reference panel comprises a relatively large set of whole-genome sequences (32,470 individuals), although it is predominantly composed of individuals of European ancestry. In our study, imputation was conducted using a reference panel that included all participants in the HRC, encompassing both mixed and non-European ancestry individuals. We completely agree about the importance of population-specific reference panels. Consequently, we have incorporated this consideration into the third limitation of the manuscript and referenced the paper mentioned by the reviewer.

[Added to the Discussion, page 15, lines 381–384]

Third, GWAS based on SNP array has limited ability to identify rare variants and population-specific signals. The utilization of population-specific reference panels for imputation might improve the discovery of additional loci⁴⁴.

5) There is no replication sample. That's unavoidable but a major limitation for the newly reported loci and should be mentioned.

Response: We sincerely appreciate the reviewers' concern regarding the critical limitation related to the absence of direct replication for the novel genetic loci discovered in the GWAS. As we acknowledge this concern, we have conducted a comprehensive replication study with independent individuals for the replication phase. The results and discussion of the replication study are summarized at the beginning of this response letter and have been incorporated into the revised manuscript.

6) Can you analyze the X chromosome? I understand the sample is mostly female (though I couldn't see a breakdown). I am sure there are some pigmentation genes on the X and also interesting in the light of sex differences in pigmentation that are mentioned.

Response: We appreciate the reviewer's insightful suggestion. We have examined the associations between genetic variants on chromosome X and skin color in female participants (**Supplementary Fig. S7**). Despite our expectations aligning with those of the reviewer, none of the tested variants showed statistical significance. To investigate the X chromosome, increased statistical power of discovery is needed.

Supplementary Fig. S7

[Added to the Results, page 5, lines 107–109]

A total of 138,839 variants on chromosome X were also tested for associations with skin color in 42,770 female participants. However, none of the tested variants exhibited statistical significance (Supplementary Fig. S7).

■ Comments about other analyses:

8) Residual stratification in the GWAS results can result in spurious Qx scores (Berg et al. eLife 2020, Sohail et al. eLife 2020), and could bias the selection results.

Response: We appreciate the reviewer’s comment. Because a Bayesian linear mixed model (BOLT-LMM) with PCs as covariates was applied for the association test, the residual stratification was adequately calibrated. The effect size estimates in the discovery GWAS were not correlated with the variant loadings of the first 10 PCs (Supplementary Fig. S6).

Supplementary Fig. S6

[Added to the Results, page 5, lines 103–105]

The effect size estimates in the discovery GWAS were not correlated with the variant loadings of the first 10 principal components that represent the population structure (Supplementary Fig. S6).

9) The P-values for the correlations between genetic score and latitude/solar radiation are not correct – the points are not independent so you cannot use the t-test (you need a mantel test or similar).

Response: We appreciate the reviewer’s statistical comment and suggestion. The P -value of Spearman’s correlation coefficient was estimated under the null distribution of all possible permutations. As pointed out by the reviewer, we recognize the inaccurate P -values due to the correlation within populations in the same continent. Most of the Mantel test results did not reach statistical significance, probably due to the lack of power due to the limited number of data points. Therefore, we have revised the sentence that imply a strong correlation and have included the P -value of the Mantel test in the revised manuscript.

[Added to the Results, page 10, lines 244–245]

Genetic scores for individual populations showed correlation with geographic and environmental factors, albeit with P -values estimated from the Mantel test (P_{mantel}) generally being underpowered.

[Added to the Results, page 10, lines 248–255]

The genetic score for L^* was positively and negatively correlated with absolute latitude ($r_s = 0.513$, $P_{\text{permutation}} = 6.90 \times 10^{-3}$, $P_{\text{Mantel}} = 0.160$) and mean annual solar radiation ($r_s = -0.496$, $P_{\text{permutation}} = 9.30 \times 10^{-3}$, $P_{\text{Mantel}} = 0.130$), respectively. Genetic scores for a^* and b^* were negatively and positively correlated with absolute latitude (a^* , $r_s = -0.581$, $P_{\text{permutation}} = 1.79 \times 10^{-3}$, $P_{\text{Mantel}} = 7.80 \times 10^{-3}$; b^* , $r_s = -0.562$, $P_{\text{permutation}} = 2.70 \times 10^{-3}$, $P_{\text{Mantel}} = 0.072$) and mean annual solar radiation (a^* , $r_s = 0.547$, $P_{\text{permutation}} = 3.63 \times 10^{-3}$, $P_{\text{Mantel}} = 0.121$; b^* , $r_s = 0.526$, $P_{\text{permutation}} = 5.40 \times 10^{-3}$, $P_{\text{Mantel}} = 0.166$), respectively.

[Added to the Methods, page 23, lines 609–612]

Spearman’s correlation coefficient (r_s) was used as a measure of correlation. To assess the significance of correlation, the P -value of r_s was estimated under the null distribution of all possible permutations ($P_{\text{permutation}}$) and by the Mantel test (P_{Mantel}).

10) In Figure 4a there seem to be three sizes of circle on the plot, but only two described in the legend.

Response: We appreciate the reviewer’s careful review. We have added the omitted legend for the size of circles in **Fig. 4a** and **Supplementary Fig. S21a**.

11) How did you decide the latitudes for the 1000 Genomes populations (STU & and ITU are from the UK, latitude ~50N and CEU are from Utah, latitude ~30). I get that you are trying to local the “ancestral” populations – please just explain how you did so.

Response: We appreciate the reviewer’s comment on the description of the latitudes. As described in the reviewer’s comment, we tried to investigate the geographical location that represents the ancestral population. We approximated the longitude and latitude of each ancestral population based on the geographic region in which the population was investigated by the 1000 Genomes Project, similar to the study by Adhikari et al. (Nature Communications, 2019).

[Added to the Methods, page 23, lines 606–608]

The longitude and latitude of each population were approximated based on the geographic region in which the population was investigated, similar to the study by Adhikari *et al*⁸.

12) I think it would be good to integrate the functional evidence from melanocytes in this study: <https://www.science.org/doi/10.1126/science.ade6289>. How many of your novel genes were identified here.

Response: We appreciate the reviewer's suggestion of the recent paper on human pigmentation. Nine of our colocalized or nearest genes listed in Fig. 3c were compared to the results of that paper. Of these, three previously reported genes were included in the 169 putative melanin-promoting genes in the referred paper, and the rest, including three previously unreported genes (*CTR9*, *PM20D1*, and *SLC45A3*), were not found in the referred paper. Differences in the ethnicity of participants between our study (East Asian) and the referred paper (white British), as well as differences in whether genes were observed within large populations, may have limited the ability to integrate functional evidence.

Thank you very much for the valuable time and effort that you have put into reviewing our manuscript.

Reviewer #2:

■ Remarks to the Author:

This manuscript describes the results of a GWAS on skin colour in East Asians together with some additional types of data analyses on the discovered genetic loci. I agree with the authors that this is an interesting study because skin colour GWASs have not yet been carried out on the level of study population diversity that the global distribution of skin colour diversity would expect, and thus what is needed to genetically understand the full range of skin colour in humans, for which global populations need to be involved in GWAS. Study sample size is with 48,500 not very high, but acceptable given the scarcity of skin colour GWASs in Asians. Seeing image-based, continuous skin colour phenotype measures being used for GWAS is appreciated. This reviewer has two major conceptual concerns and some minor technical concerns as outlined in the following.

■ Major conceptual concern:

1) The most concerned limitation of this study is the lack of direct replication of the novel genetic loci the authors discovered in their GWAS. Although the authors compared the effect of their lead SNPs in their study population with those in East Asians from UKBB and report “comparable” effects, UKBB East Asians are with 2300 small in number and the reported correlation of the effects is with $r=0.83$ not very strong, thus providing limited indirect replication evidence. Moreover, the authors do not present any direct replication evidence of their novel SNPs, not in UKBB East Asians nor in any other East Asian population data. In general, direct replication of novel discovery findings is common standard in GWASs. Any reader would be reluctant to consider the newly presented skin colour loci as true genetic determinants of skin colour without direct replication evidence being presented. Moreover, this reviewer would expect that for a publication in a journal of the high reputation as this one, direct replication evidence would be a prime prerequisite for publishing GWASs. Why did the authors not use UKBB East Asians for direct replication testing? Why did the authors not leave out from their discovery GWAS dataset some of their samples and use them for direct replication testing? Although the latter would represent internal replication testing as such samples come from the same pool, together with UKBB direct replication

results from a relatively small sample set, the combination of both direct replication efforts would give a better idea on replication of their novel findings than the indirect analyses that allow limited conclusions they have performed thus far in UKBB.

Response: We sincerely appreciate the reviewer's positive remark on the value of our study and constructive comments. As we acknowledge the concern regarding the lack of direct replication, we have conducted a comprehensive replication study with independent individuals for the replication phase. The results and discussion of the replication study are summarized at the beginning of this response letter and included into the revised manuscript.

2) South Koreans used here are not widely known as expressing large i.e., categorical skin colour differences as known from other Asian countries such as India. Thus, the reader wonders how the measured quantitative skin colour variation in the Koreans used here compares to that of populations with large categorical skin colour differences. If such comparative data are not available, it would be interesting to know how the measured quantitative skin colour variation translates to variation in commonly used skin colour categories. Are all Korean samples tested fall into one skin colour category or more categories? Such information may allow understanding why a sample size of close to 50,000 did not allow finding more than 23 associated genetic loci. Maybe the answer lies in the limited skin colour diversity of the study population, which also is a problem when carrying out skin colour GWASs in Europeans and in most samples collected within a single country that share similar genetic ancestry. The reported explained variance values make it clear that the number of skin colour genes in East Asians will be very high and most remain uncovered by this study. This may be due to their extremely small effect size not detectable with the statistical power available with 50,000 samples, or by the low phenotypic diversity of the study population, or both. Shedding more light into this important aspect would be import for interpreting the authors' findings. Therefore, additional analyses on the phenotypic diversity of the study population would be highly appreciated.

Response: We really appreciate the reviewer's valuable advice. We agree that further analyses

of the phenotypic diversity of the study population is warranted. We have revised **Fig. 1** to provide a clearer description of phenotype distribution. To assess the diversity of skin color in this study, we categorized the skin color measurement indices (CIE LAB values) according to the ITA^o criteria, a globally recognized standard for skin color classification (Fig. 1c). The study population fell into five categories, with the majority falling into the tan (64.8%), intermediate (20.9%), and brown (12.8%) categories. As mentioned by the reviewer and in our manuscript, skin color in East Asians is a phenotype with relatively low diversity compared to populations worldwide (page 14, lines 149–151). Despite this limitation, our study identified several potential causal loci including previously unreported ones. We suggest that the reason our findings might be attributed to the relatively large sample size and objectively quantified continuous skin color measurements. To substantiate our assertion, we have conducted a GWAS of categorized skin color (5 categories as GWAS outcome) using POLMM (**Supplementary Fig. S9**). In the categorical GWAS, only 11 of 26 lead variants were identified with no additional loci showing significant associated loci.

Fig. 1.

Supplementary Fig. S9.

[Added to the Results, page 5, lines 118–122]

This GWAS of objectively quantified skin color traits identified more significant loci than a GWAS based on the categorized skin color according to the individual typology angle (ITA°) value criteria (Fig. 1c and Supplementary Fig. S9). The GWAS of the categorical skin color using POLMM¹⁶ identified 11 of 26 lead variants, with no additional significant loci, including two and nine variants in previously unreported and reported loci, respectively.

[Added to the Discussion, page 12, lines 310–312]

This GWAS of objectively quantified skin color traits (CIE LAB values; L^* , a^* , and b^*) produced more powerful results compared to GWAS based on ITA° value or questionnaire-based categorical skin color (Supplementary Table S1).

[Added to the Methods, page 18, lines 477–481]

Genome-wide associations between variants and categorical skin color were tested using POLMM¹⁶, a proportional odds logistic mixed model. Skin color of the study participants was categorized with criteria of individual typology angle (ITA°) value⁵⁰, which has been used globally for skin color classification, to generate the categorical skin color. ITA° was calculated according to the following equation: $ITA^\circ = \left[\text{ArcTan} \left(\frac{L^* - 50}{b^*} \right) \right] \times \frac{180}{\pi}$.

■ Minor technical concerns:

3) The authors report different numbers of identified significantly associated genetic loci with the different quantitative measures of skin colour they used without discussing these differences. It would be interesting to know why, in the view of the authors, they obtained such different results.

Response: We appreciate the valuable comment from the reviewer. We have included additional discussion with regards to the varied significance depending on quantitative measures of skin color.

[Added to the Discussion, page 13, lines 327–333]

Skin color can be subdivided into brightness, redness, and yellowness, which can be represented by CIE LAB values, and is influenced by factors such as melanin, hemoglobin, oxyhemoglobin, and carotenoid levels²². The skin color-associated loci identified in this study exhibited varying significance depending on the CIE LAB value, implying that these regions may be influenced by different factors and contribute to distinct biological pathways. For example, *MFSD12* contributes to red-yellow pigmentation by maintaining cysteine levels within melanosomes^{23,24}, and *SCARB1* influences skin yellowness by promoting the uptake of carotenoids^{25,26}.

4) Why was heritability testing done in an age-group stratified way? What is the authors' explanation that highest heritability was found in the youngest age group?

Response: We appreciate the reviewer's comment on the estimation of heritability stratified by age group. To test our hypothesis that different age groups may have different levels of exposure to environmental factors on skin color (for example, one group may be exposed to less environmental effects and show higher heritability than another group), we conducted the stratified estimation by age groups. To further clarify our hypothesis, we analyzed the distribution of the sun exposure variable across different age groups (**Supplementary Fig. S2**) and estimated the impact of sun exposure on skin color in each age group (**Supplementary Table S3**).

Supplementary Fig. S2.

Summary of Supplementary Table S3.

Phenotype	Sun exposure	Age group	β	s.e.	L95	U95	P
L^*	Sun exposure time per day	old group (> 49 years)	0.203	0.063	0.080	0.326	1.22E-03
L^*	Sunblock usage	old group (> 49 years)	0.155	0.069	0.020	0.289	0.024
b^*	Sun exposure time per day	middle group (37-49)	0.275	0.050	0.176	0.374	4.84E-08
b^*	Sun exposure time per day	old group (> 49 years)	0.135	0.052	0.034	0.237	8.74E-03
b^*	Sunblock usage	middle group (37-49)	0.136	0.056	0.027	0.245	0.015

[Added to the Results, page 4, lines 87–93]

The distribution and effect of sun exposure variables varied across age groups (young [< 37 years], middle [37–49 years], and old [> 49 years] age groups) (Supplementary Fig. 2 and Supplementary Table S3). For example, the effect size of sun exposure time per day and sunblock usage on L^* was 1.50 (effect size of the interaction term between sun exposure variable and age group [$\beta_{\text{sun} \times \text{age}}$] = 0.203, $P = 1.22 \times 10^{-3}$) and 1.48 ($\beta_{\text{sun} \times \text{age}} = 0.155$, $P = 0.024$) times greater, respectively, in the old age group when compared with the young age group.

[Added to the Results, pages 7–8, lines 177–179]

In consideration of the observation that the effect of the environment varied across age groups, we also estimated SNP-based heritability in the three female age groups (Fig. 2c and Supplementary Table S9).

[Added to the Discussion, pages 13, lines 316–319]

The highest SNP-based heritability was estimated in the youngest age group (41.2%, 31.0%, and 30.7% for L^* , a^* , and b^* , respectively, in females younger than 37 years), presumably due to their reduced environmental influence on skin color compared to other age groups.

[Added to the Methods, page 20, lines 527–534]

To evaluate the discrepancy of sun exposure variables and their effects on skin color across age groups, we conducted linear regressions in 36,246 independent female participants. The associations between age group and sun exposure variable were tested to assess the variation of sun exposure variable across age groups. The interplay between each sun exposure variable and age group on skin color were examined to assess the discrepancy in the effects of sun exposure variable on skin color across age groups. These regression models included covariates such as measurement month, camera resolution, genotyping batches, and the first 10 PCs of genetic ancestry.

5) In the methods section it says that >52,000 samples were included in this study, while only 48,500 were used in the GWAS. How were the samples used for GWAS selected from the total samples? Why were 3,500 samples excluded?

Response: We appreciate the reviewer's comment. Prior to genotyping, 3,433 individuals with any unreliable phenotype were excluded. Additionally, 846 genotyped individuals were excluded during the sample quality control (251 individuals with call rate < 95%, 319 individuals with extreme heterozygosity rate, and 276 individuals with discordance between the reported and inferred sex). We have included this detailed information in the Methods section and Supplementary Note.

[Added to the Methods, page 17, lines 436–439]

To obtain reliable results from this study, we excluded participants with 1) measured images with low quality via manual curation and 2) the self-reported items that were possibly entered incorrectly (height below 1 m or above 2.5 m; weight below 30 kg or above 200 kg; age under 10 years). In total, 49,279 East Asians were genotyped.

[Added to the Supplementary Note, page 4, lines 78–80]

Samples were excluded based on the following criteria: call rate < 95% (251 individuals were excluded), heterozygosity rate three standard deviations away from the mean (319 individuals were excluded), and discordance between the reported and inferred sex based on the heterozygosity rate on chromosome X (276 individuals were excluded).

6) The authors write that polygenic scores were estimated from 40,790 unrelated study participants. Does this imply that the 48,500 individuals used for discovery GWAS include 8000 relatives? I cannot see how such relatedness effects were adjusted for in the discovery GWAS.

Response: We appreciate this comment. To adjust for the relatedness effect and population stratifications, we applied a Bayesian linear mixed model (BOLT-LMM) with PCs as covariates, for the association test. BOLT-LMM is a commonly used method for adjusting both relatedness effect and population stratifications. We have clarified the statements in the manuscript to avoid possible confusion to readers.

[Added to the Results, page 5, lines 101–103]

By applying a Bayesian linear mixed model (BOLT-LMM)¹² with PCs as covariates, no evidence of population stratification was observed in quantile–quantile plots of the GWAS results (Supplementary Fig. S5).

[Added to the Methods, page 18, lines 458–459]

Associations between variants and skin color were tested using BOLT-LMM (v.2.3.4)¹², a Bayesian linear mixed model, to adjust for sample relationships.

7) What does “15, 15, and 13 variants at 13 loci for L^* ” in line 100 mean? Elsewhere in the manuscript, they refer to 15 L^* lead variants.

Response: We appreciate the reviewer’s careful attention to the sentence. We have clarified the sentence to avoid confusion.

[Added to the Results, page 5, lines 110–112]

We identified 26 lead variants at 23 independent loci associated with skin color traits, including 15 variants at 13 loci for L^* , 15 variants at 13 loci for a^* , and 13 variants at 12 loci for b^* , respectively (Table 1 and Supplementary Table S4).

8) The effect comparison with East Asian UKBB was done for 15 L^* lead variants. Why only for these and not for all novel lead variants they discovered?

Response: We appreciate the reviewer's comment. The skin color phenotype analyzed in the UKBB (data field 1717) is limited to brightness (six categories), representing a categorical version of L^* . Consequently, it was considered more appropriate to compare across ethnic groups by using the L^* lead variants. We have added a sentence to provide clarification on the reason for comparing L^* lead variants rather than all lead variants.

[Added to the Results, page 11, lines 264–265]

The skin color phenotype analyzed in the UKBB was limited to brightness, representing a categorical version of L^* .

9) From their GWAS, the authors report 23 skin colour association genetic loci, while in their gene expression analysis they tested and provide results for 36 skin colour associated genes. How come the larger number of genes used in the gene expression analysis?

Response: We appreciate the reviewer's careful comment on the confusion. We identified a total of 41 skin color-associated genes, including 23 nearest and 18 colocalized genes, listed in **Table 1**. In the scRNA-seq data, 36 out of the 41 identified skin color-associated genes were available. We have added a sentence to the Results section to provide clarification on the analysis process.

[Added to the Results, page 9, lines 217–218]

In the integrated scRNA-seq data, 36 out of the 41 identified skin color-associated genes, including both the nearest and colocalized genes, were available.

10) The UK Biobank study is typically abbreviated as UKBB, not as UKB used by the authors.

Response: We thank the reviewer for the comment. We have modified the abbreviation for the UK Biobank from ‘UKB’ to ‘UKBB’ throughout the manuscript, Figures, and Tables.

We sincerely appreciate the valuable time and effort that you have dedicated to reviewing our manuscript.

REVIEWERS' COMMENTS

Reviewer #1 (Remarks to the Author):

I appreciate the work the authors have done on the revisions. I think the replication analysis is reasonable given the limitations of data collection, and provides some additional support for the novel loci. I remain highly enthusiastic about the paper and have only a few minor comments remaining.

1) Data availability: the link is just a generic link to the gwas catalog. The authors need to provide an accession number (you can obtain this now even if you embargo the results until the paper is published). I would want to be sure that the full summary statistics are available; i.e. every SNP for all three of the phenotypes, not only the significant SNPs.

2) RE conflicts of interest; my understanding is that employment or stock ownership in a company that may benefit financially from the publication would be declared. In my opinion there could be a potential benefit here, but I leave it up to the editors. Usually I just see a note like "these authors are employees or hold stock in xx company".

3) Line 222; This is confusing because the 12 genes listed here are referring to the 12 genes mentioned on line 220, not the 5 of 12 genes listed on line 221. I suggest re-ordering this sentence.

4) Figure 1; Are the skin tones in Figure 1 a and c accurate? They look quite dark for this population. Perhaps it's a display issue?

5) I assume the genotype/phenotype data are not available. If so please make that clear.

Reviewer #2 (Remarks to the Author):

I am pleased to see the efforts the authors placed into addressing my previous comments in their revised manuscript, including the replication efforts they previously missed completely and have now provided. However, as expected based on the large difference in the number of samples the authors used in their discovery GWAS and what they used for replication, with the latter being about 10x smaller than the former, the replication results are not providing convincing evidence in support of the newly discovered loci, which anyways are not more than 11. I leave it up to the editors to decide whether the reported quantity level of novelty and the achieved level of trust that the newly reported loci are true positives, which both are limited as far as I see it, is sufficient for this Journal's publication standards for GWAS papers.

Reviewer #4 (Remarks to the Author):

The paper is of interest.

The authors provide reply to critique in separate letter.

Unfortunately, it is difficult to judge to which degree these changes were included, since the changes are not parked in the submitted pdf file. This is a serious oversight.

The introduction is suboptimal with some deficiencies in photobiology and vitamin D

For example, it is UVB but not other wavelengths of UVR that are responsible for vitamin D formation.

Please indicate this and cite one of Holick's papers.

As relates to biological action of UVR, the authors should place them in broader context with referral to (Endocrinology 159(5), 1992-2007, 2018). PMID: PMC5905393).

The discussion indicates some deficiencies in mechanisms regulating melanin pigmentation, which

have to be corrected. For example, in the skin it is regulated by complex cutaneous neuroendocrine system that respond to UVR (American Journal of Physiology-Cell Physiology 323:6, C1757-C1776, 2022 . (DOI: 10.1152/ajpcell.00147.2022).

Also, the readers would appreciate some basic information on biochemistry and chemistry of melanogenesis and its regulation (Frontiers in Oncology 2022;12. DOI: 10.3389/fonc.2022.842496). Finally, some references on above topics are random and not representative. This has to be corrected. There are spelling or stylistic errors that have to be corrected.

Responses from the Authors to Review Comments:

We thank the editor and reviewers for their constructive comments and suggestions. We have revised our manuscript to address them. We sincerely appreciate the valuable time and effort that you have dedicated to reviewing our manuscript.

REVIEWER COMMENTS:

Reviewer #1:

■ Remarks to the Author:

I appreciate the work the authors have done on the revisions. I think the replication analysis is reasonable given the limitations of data collection, and provides some additional support for the novel loci. I remain highly enthusiastic about the paper and have only a few minor comments remaining.

Response: We appreciate the reviewer's positive evaluation of our work. As we fully acknowledge the reviewer's concerns, we have revised our manuscript.

1) Data availability: the link is just a generic link to the gwas catalog. The authors need to provide an accession number (you can obtain this now even if you embargo the results until the paper is published). I would want to be sure that the full summary statistics are available; i.e. every SNP for all three of the phenotypes, not only the significant SNPs.

Response: We appreciate the reviewer's comment. We have submitted the full summary statistics to the GWAS Catalog and provided the accession numbers in the revised manuscript.

[Added to the Data Availability, page 25, lines 643–650]

The full summary statistics of GWAS for L^* (luminance), a^* (red/green component), and b^*

(yellow/blue component) in 48,433 East Asians are publicly available at the NHGRI-EBI GWAS Catalog (<https://www.ebi.ac.uk/gwas/downloads>) with accession numbers GCST90320257, GCST90320258, and GCST90320259, respectively. The summary statistics of associations of variants in chromosome X with L^* (luminance), a^* (red/green component), and b^* (yellow/blue component) in 42,770 East Asian females are publicly available at the NHGRI-EBI GWAS Catalog with accession numbers GCST90320260, GCST90320261, and GCST90320262, respectively.

2) RE conflicts of interest; my understanding is that employment or stock ownership in a company that may benefit financially from the publication would be declared. In my opinion there could be a potential benefit here, but I leave it up to the editors. Usually I just see a note like “these authors are employees or hold stock in xx company”.

Response: We appreciate the reviewer’s concern regarding the competing interests. We have added a competing interest statement for employment.

[Added to the Competing Interests Statement, page 32, lines 831–834]

Migenstory's business is exclusively involved in providing Direct-to-Consumer (DTC) genetic testing services and generating data for research at LG H&H, without any engagement in the development of medicine or related technologies. J.G.Shin, S.Leem, H.Kim, K.N.Gu, S.W.You, S.G.Park, Y.Kim, and N.G.Kang are employees of LG H&H. Other authors declare no competing interests.

3) Line 222; This is confusing because the 12 genes listed here are referring to the 12 genes mentioned on line 220, not the 5 of 12 genes listed on line 221. I suggest re-ordering this sentence.

Response: We sincerely appreciate the reviewer’s careful attention to the sentence. As the reviewer's recommendation, we have clarified the sentence to avoid confusion. Furthermore, we have identified an error in the number of genes in previously unreported loci among genes

exhibited the highest expression levels in melanocytes. This error has been corrected as follows.

[Revised to the Results, page 9, lines 221–225]

At the individual gene level, one-third of the tested skin color-associated genes (12 of 36 genes) exhibited the highest expression levels in melanocytes (Fig. 3c and Supplementary Table S12), and 42.33% (5 of 12 genes; *USP35*, *BCKDHB*, *GAB2*, and *MRPS22*) were in previously unreported loci: *MFS12*, *MC1R*, *OCA2*, *RAB32*, *USP35*, *BCKDHB*, *SLC6A17*, *BNC2*, *DEF8*, *GAB2*, and *SNORC*, and *MRPS22* were in previously reported loci.

4) Figure 1; Are the skin tones in Figure 1 a and c accurate? They look quite dark for this population. Perhaps it's a display issue?

Response: We appreciate the reviewer's comment. Each dot in **Fig. 1** represents an accurate reflection of the measured skin color. It may appear darker than expected due to the utilization of an average value of the color of both cheek areas, rather than localized measurements. We have revised a sentence in the legend of **Fig. 1** to provide clarification.

[Revised to the legend of Fig. 1, page 34, lines 844–845]

Each dot corresponds to a study participant and its color represents an average value of the measured skin color of both cheek areas for that person.

5) I assume the genotype/phenotype data are not available. If so please make that clear.

Response: We appreciate the reviewer's comment. Unfortunately, we are unable to release individual-level genotype and phenotype data because study participants did not consent to provide their personal information to the public. We have added a sentence to provide clarification on the data availability.

[Added to the Data Availability, page 25, lines 641–643]

This individual-level genotype and phenotype data are protected and are not available due to data privacy laws.

Reviewer #2:

■ Remarks to the Author:

I am pleased to see the efforts the authors placed into addressing my previous comments in their revised manuscript, including the replication efforts they previously missed completely and have now provided. However, as expected based on the large difference in the number of samples the authors used in their discovery GWAS and what they used for replication, with the latter being about 10x smaller than the former, the replication results are not providing convincing evidence in support of the newly discovered loci, which anyways are not more than 11. I leave it up to the editors to decide whether the reported quantity level of novelty and the achieved level of trust that the newly reported loci are true positives, which both are limited as far as I see it, is sufficient for this Journal's publication standards for GWAS papers.

Response: We appreciate the reviewer's positive evaluation of our replication efforts. As we fully acknowledge the reviewer's concerns, we have revised our manuscript. As mentioned by the reviewer and in our manuscript, the association of the lead variants derived from the GWAS was partially replicated in 10% of individuals of the discovery cohort. Despite this limitation, this study included the largest sample size in non-European populations to date, resulting in increased statistical power compared with previous studies. We hope that GWAS will be performed in diverse populations and trans-ancestry meta-analyses will be conducted in order to identify population-specific or shared genetic factors affecting skin color.

Reviewer #4:

■ Remarks to the Author:

The paper is of interest.

The authors provide reply to critique in separate letter.

Unfortunately, it is difficult to judge to which degree these changes were included, since the changes are not parked in the submitted pdf file. This is a serious oversight.

The introduction is suboptimal with some deficiencies in photobiology and vitamin D

For example, it is UVB but not other wavelengths of UVR that are responsible for vitamin D formation. Please indicate this and cite one of Holick's papers.

As relates to biological action of UVR, the authors should place them in broader context with referral to (Endocrinology 159(5), 1992-2007, 2018). PMID: 3005393).

The discussion indicates some deficiencies in mechanisms regulating melanin pigmentation, which have to be corrected. For example, in the skin it is regulated by complex cutaneous neuroendocrine system that respond to UVR (American Journal of Physiology-Cell Physiology 323:6, C1757-C1776, 2022 . (DOI: 10.1152/ajpcell.00147.2022).

Also, the readers would appreciate some basic information on biochemistry and chemistry of melanogenesis and its regulation (Frontiers in Oncology 2022;12. DOI: 10.3389/fonc.2022.842496).

Finally, some references on above topics are random and not representative. This has to be corrected.

There are spelling or stylistic errors that have to be corrected.

Response: We appreciate the reviewer's positive evaluation of our work and valuable advice. As we fully acknowledge the reviewer's concerns and constructive suggestions, we have revised our manuscript.

The introduction is suboptimal with some deficiencies in photobiology and vitamin D

For example, it is UVB but not other wavelengths of UVR that are responsible for vitamin D formation. Please indicate this and cite one of Holick's papers.

As relates to biological action of UVR, the authors should place them in broader context

with referral to (Endocrinology 159(5), 1992-2007, 2018). PMID: PMC5905393).

Response: We appreciate the reviewer's careful recommendation. We have revised and added sentences related to the vitamin D synthesis with the appropriate references.

[Revised to the Introduction, page 3, lines 45–46]

..., whereas at higher latitudes, light skin is advantageous for vitamin D synthesis in reduced UV-B light.

[Added to the Introduction, page 3, lines 48–49]

The UV-B is responsible for vitamin D formation and affect endocrine gland functions and overall body homeostasis^{6,7}.

[Revised to the Discussion, page 14, lines 347–348]

It is well known that UV-B exposure is a key factor in vitamin D synthesis⁶.

[Added to the References, page 26, lines 676–679]

6. Holick, M.F. Vitamin D deficiency. *N Engl J Med* **357**, 266-81 (2007).
7. Slominski, A.T., Zmijewski, M.A., Plonka, P.M., Szaflarski, J.P. & Paus, R. How UV Light Touches the Brain and Endocrine System Through Skin, and Why. *Endocrinology* **159**, 1992-2007 (2018).

The discussion indicates some deficiencies in mechanisms regulating melanin pigmentation, which have to be corrected. For example, in the skin it is regulated by complex cutaneous neuroendocrine system that respond to UVR (American Journal of Physiology-Cell Physiology 323:6, C1757-C1776, 2022 . (DOI: 10.1152/ajpcell.00147.2022).

Also, the readers would appreciate some basic information on biochemistry and chemistry of melanogenesis and its regulation (Frontiers in Oncology 2022;12. DOI: 10.3389/fonc.2022.842496).

Response: We appreciate the reviewer's insightful comment. We have added the information on biochemistry and chemistry of melanogenesis in a broad context with related papers in the discussion as suggested by the reviewer.

[Added to the Discussion, page 13, lines 330–332]

Melanin serves as a primary factor in protecting the skin from UV radiation and its regulation within cells involves intricate mechanisms mediated by various endocrine and biochemical signaling pathways^{25,26}.

[Added to the References, pages 27–28, lines 717–720]

25. Slominski, A.T. et al. Neuroendocrine signaling in the skin with a special focus on the epidermal neuropeptides. *Am J Physiol Cell Physiol* **323**, C1757-C1776 (2022).
26. Slominski, R.M. et al. Melanoma, Melanin, and Melanogenesis: The Yin and Yang Relationship. *Front Oncol* **12**, 842496 (2022).